# Extreme coastal water level estimation and projection: A comparison of statistical methods

Maria Francesca Caruso[1] and Marco Marani[1,2]

[1]Department of Civil, Architectural, and Environmental Engineering, University of Padova, 35131, Padova, Italy
[2]Nicholas School of the Environment and Department of Civil and Environmental Engineering, Duke University, Durham, NC, USA

**Correspondence:** Caruso, M.F. (mariafrancesca.caruso@phd.unipd.it)

**Abstract.** Accurate estimates of the probability of extreme sea levels are pivotal for assessing risk and for designing coastal defense structures. This probability is typically estimated by modelling observed sea-level records using one of a few statistical approaches. In this study we comparatively apply the Generalized Extreme Value (GEV) distribution, based on Block Maxima (BM) and Peaks-Over-Threshold (POT) formulations, and the recent Metastatistical Extreme Value Distribution (MEVD) to four long time series of sea-level observations distributed along European coastlines. A cross-validation approach, dividing available data into separate calibration and test sub-samples, is used to compare their performances in high-quantile estimation. To address the limitations posed by the length of the observational time series, we quantify the estimation uncertainty associated with different calibration sample sizes, from 5 to 30 years. We study extreme values of the coastal water level – the sum of the water level setup induced by meteorological forcing and of the astronomical tide – and we find that the MEVD framework provides robust quantile estimates, especially when longer sample sizes of 10-30 years are considered. However, differences in performance among the approaches explored are subtle, and a definitive conclusion on an optimal solution independent of the return period of interest remains elusive. Finally, we investigate the influence of end-of-century projected mean sea levels, on the probability of occurrence of extreme total water levels (the sum of the instantaneous water level and the increasing mean sea level) frequencies. The analyses show that increases in the value of total water levels corresponding to a fixed return period are highly heterogeneous across the locations explored.

## 1 Introduction

The statistical analysis of extreme values of random variables is of wide conceptual and applicative importance in science and engineering (Coles, 2001; Beirlant et al., 2004; Castillo et al., 2005; Finkenstädt and Rootzén, 2004). Modelling extreme value probability of occurrence is indeed an active field of theoretical and applied research in many fields, such as hydrology and climatology (Katz et al., 2002; Cancelliere, 2017; Mekonnen et al., 2021; Miniussi and Marra, 2021), ecology (Katz et al., 2005; Rypkema et al., 2019), ocean wave modelling (Rueda et al., 2016; Benetazzo et al., 2017; Barbariol et al., 2019), transport engineering (Songchitruksa and Tarko, 2006), geophysical processes (Pisarenko et al., 2014a, b; Elvidge and Angling, 2018; Hosseini et al., 2020), biomedical data analysis (De Zea Bermudez and Mendes, 2012; Chiu et al., 2018), insurance and financial applications (Embrechts et al., 1997; Chan et al., 2022), and many others.

In particular, the reliable estimation of the occurrence probability of coastal flooding events of large magnitude is crucial to environmental hazard evaluation (Coles and Tawn, 2005; Hamdi et al., 2018) and to decision-making and mitigation measure design. In fact, coastal flooding hazard has been increasing at the global scale in recent decades, a trend expected to continue as a result of climate change (Meehl et al., 2007; Church et al., 2013; Fortunato et al., 2016). Several studies highlight that global sea-level rise will continue accelerating in the $21^{st}$ century as a consequence of climate change (Church and White, 2006; Jevrejeva et al., 2008; Church and White, 2011; Haigh et al., 2014b; Hay et al., 2015). Additionally, changes in storminess may have an important role in modifying the frequency and magnitude of water level extremes (Lowe et al., 2010; Menéndez and Woodworth, 2010; Woodworth et al., 2011). Much of the current work on extreme coastal flooding events is based on the classical Extreme Value Theory (EVT) (Fréchet, 1927; Dalrymple, 1960; Coles, 2001; Woodworth and Blackman, 2002; Hamdi et al., 2014, 2015, and references therein), which identifies the family of distribution functions known as Generalized Extreme Value (GEV) distribution (Von Mises, 1936) as a general model for the distribution of maxima (or minima) extracted from fixed time periods of equal length ("blocks", most commonly with length of one year). The GEV, according to its original formulation, arises as a limiting distribution for maxima (or minima, not considered here) of a sequence of independent and identically distributed (i.i.d.) random variables. The Peaks-Over-Threshold (POT) formulation (Balkema and de Haan, 1974; Pickands, 1975), extends the original GEV derivation by modelling all events exceeding a high threshold, as opposed to considering just yearly maxima as in the GEV-Block Maxima formulation (GEV-BM). The POT approach again recovers the GEV distribution as the distribution of the annual maxima if two assumptions are valid (Davison and Smith, 1990): 1) the number of events/year is Poisson-distributed; 2) exceedances over the threshold come from a Generalized Pareto Distribution (GPD). Under these suitable conditions, in the following we will refer to the POT framework as POT-GPD formulation. For a brief overview of the theory underlying EVT and the two main methods based on the GEV distribution (i.e. BM and POT approaches), the reader can refer to the Method section or the supplementary material. The POT-GPD approach is often considered to be superior to GEV-BM in practical applications, due to its more efficient use of often scarce observations. For extreme sea-level studies in particular, Coles and Tawn (2005) and Haigh et al. (2010) recognize two weakness in the use of the GEV-BM analysis: 1) sea level is the combination of tide-driven (deterministic) and storm-driven (stochastic) components. The presence of a deterministic component is suggested to violate the i.i.d. assumption required in the GEV-BM derivation; 2) sea-level data are collected frequently (e.g., hourly), while the GEV-BM approach only studies annual maxima, with an extremely inefficient use of the data. The POT framework exploits more of the available information with respect to the BM approach (e.g., Coles, 2001; Bernardara et al., 2014). However, the choice of a suitable threshold to retain a few above-threshold events/year is a critical step, and the estimation uncertainty significantly depends on threshold selection (Önöz and Bayazit, 2001; Li et al., 2012; Solari et al., 2017). The selected threshold value implies a balance between bias and estimation error variance (Coles, 2001). In fact, too low a threshold will violate the independence hypothesis of the framework, leading to bias, while too high a threshold will retain just a few values above the threshold, leading to high error variance.

More generally, GEV-based approaches, by construction, discard most of the observations, and do not attempt to optimize the use of the available information (Volpi et al., 2019). Furthermore, the traditional extreme value theory derives the GEV distribution either as the asymptotic distribution when the number of events/block becomes very large, or through the ad-hoc

GPD-Poisson assumptions underlying the POT approach. Whether these hypotheses do apply to the case of sea levels is a matter of discussion, but it seems beneficial to adopt methods that require the least amount of a-priori assumptions on the properties of the event arrival process. As a contribution to overcoming the limitations of the traditional EVT, here we explore the use of an alternative approach for modelling extreme sea levels, the Metastatistical Extreme Value Distribution (MEVD). This approach was introduced by Marani and Ignaccolo (2015) and has been previously applied to rainfall, flood-frequency analysis, and hurricane intensities. The MEVD models the distribution of yearly maxima starting from the distribution of "ordinary values", i.e. all the available data, in contrast to just considering annual maxima or a few values above a threshold. Moreover, the MEVD framework (i) is a non-asymptotic extreme value distribution, which does not require the number of events/year to be large as in the traditional theory, and (ii) makes no a-priori assumptions on the properties of the event occurrence process. In previous applications, the MEVD has been shown to significantly reduce estimation uncertainty compared to traditional approaches, especially when considering return periods greater than the sample size used for parameter estimation (Zorzetto et al., 2016; Marra et al., 2018; Miniussi and Marani, 2020; Miniussi et al., 2020a, b).

Here we comparatively analyze the performance of GEV-based approaches and MEVD in high-quantile estimations with application to extreme sea levels at different observation sites. The aim is to: 1) identify the statistical tool affording minimal uncertainty in the estimate of extreme sea levels with assigned probability of exceedance, and 2) model and understand how climate change will affect the extreme sea-level occurrence. To achieve these objectives, we analyze selected sea level time series along the European coastline and evaluate extreme sea level predictive uncertainty by adopting a cross-validation approach, in which calibration and test samples are kept separate and independent. Subsequently, we use the optimized estimation method to infer possible changes in coastal flooding hazard under IPCC climate change scenario RCP4.5 and RCP8.5.

The structure of the paper is as follows: Section 2 outlines the sea-level data and the methodology used in this application, results are described in Section 3, while the conclusions are given in Section 4.

## 2 Materials and Methods

### 2.1 Data

The analyses were performed using daily and hourly sea-level records from four tide gauge stations (see Table 1) distributed along European coastlines: Venice (Italy), Hornbæk (Denmark), Marseille (France), and Newlyn (United Kingdom). The study sites span a variety of geographical locations, coastal morphologies and storm regimes.

Venice sea-level data (maximum and minimum daily observations) were obtained from the "Centro Previsioni e Segnalazioni Maree" of the Venice Municipality (https://www.comune.venezia.it/it/content/centro-previsioni-e-segnalazioni-maree) for the Punta della Salute gauge station . The remaining water level data, all at the hourly scale, were downloaded from the University of Hawaii Sea Level Center (UHSLC) repository (http://uhslc.soest.hawaii.edu/data/?rq#uh745a/).

All sea-level datasets span long observational periods: 148 years for Venice, 122 years for Hornbæk, 115 years for Marseille (ca. 19 missing years) and 102 years for Newlyn.

The raw data for all stations were pre-processed to eliminate: 1) years with less than six months of water level observations,

**Table 1.** Information of sea-level data used in this application.

| Site name | Country | Location (degree, min.) | | Period | Missing years (%) | Deleted years | Number of years |
|---|---|---|---|---|---|---|---|
| | | Lat. | Long. | | | | |
| Venice | Italy | 45°25.0'N | 12°20.0'E | 1872-2019 | - | - | 148 |
| Hornbæk | Denmark | 56°06.0'N | 12°28.0'E | 1891-2012 | - | 1985 | 121 |
| Marseille | France | 43°16.7'N | 5°21.2'E | 1885-2018 | 14.2 | 1897; 1918; 1919; 1928; 1937; 1940; 1998; 2009; 2010 | 106 |
| Newlyn | United Kingdom | 50°06.1'N | 5°32.5'W | 1915-2016 | - | 1984; 2010 | 100 |

and 2) days with less than 24 h of data (for the case of hourly data). This process yields four "cleaned up" time series that were subsequently used in the analyses (see Table 1). Figure 1 shows daily maximum sea levels at the gauge stations explored after pre-processing.

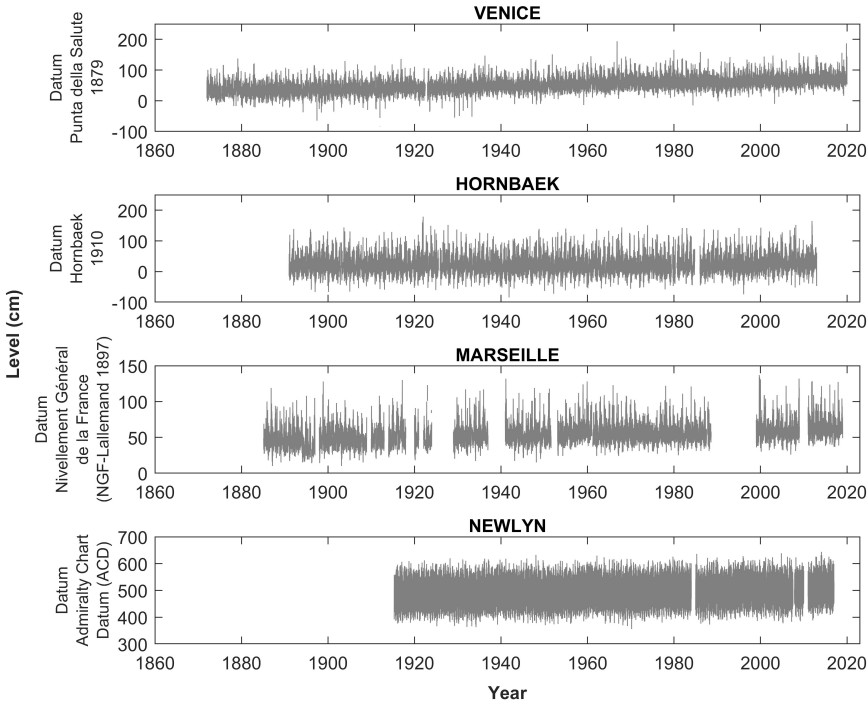

**Figure 1.** Daily maximum sea levels at different gauge stations explored after pre-processing: Venice (IT), Hornbæk (DK), Marseille (FR), and Newlyn (UK).

95

## 2.2 Methods

### 2.2.1 Mean sea level removal

The sea-level sequence is highly correlated and is generated by a non-stationary process due to long term trends in mean sea level, the deterministic tidal component, surge seasonality, and interactions between the tide and surge (Dixon and Tawn, 1999). Tide-surge interactions may change amplitude and phase of the surges, mostly in shallow estuarine areas (Johns and Ali, 1980; Bernier and Thompson, 2007; Zhang et al., 2010). Therefore, this effect needs to be taken into account when separating the surge and tide components. However, here, we do not attempt to separate these contributions, but we only analyze the sum given by the combination of the water level setup, induced by meteorological forcing, and the astronomical tide. Hence, we simply study such sum as the final result of the non-linear interactions between individual components. Under this premise, for a given site and at any instant of time $t$, the observed sea level $z(t)$ (after averaging out waves), can be split into three components (Pugh and Vassie, 1979), mean sea level, $msl(t)$, astronomically induced tidal level, $x(t)$, and meteorologically induced surge level, $y(t)$:

$$z(t) = msl(t) + x(t) + y(t) \tag{1}$$

$msl(t)$ represents the long-term variations of water levels and of the elevation datum (i.e. possible land subsidence or up-lift). Local $msl(t)$ does not change uniformly over time and its calculation is affected by many factors, such as tidal phases, long-term wind and atmospheric pressure patterns, vertical land motion (subsidence or uplift). The tidal contribution to the instantaneous sea level, $x(t)$, caused by the gravitational forces exerted by the moon and the sun is deterministic in nature, and can be predicted with a good degree of accuracy. This tidal variability occurs with characteristic periodicities between 12 hours and 18.61 years (Eliot, 2010; Haigh et al., 2011; Pugh and Woodworth, 2014; Peng et al., 2019; Valle-Levinson et al., 2021). This latter longest tidal periodicity corresponds to the precession of the lunar nodal cycle. The storm-surge contribution, $y(t)$, is the meteorologically-induced change in the water level generated by a combination of factors, such as the magnitude and direction of the wind, spatial gradients in atmospheric pressure, storm size, fetch, bathymetry, storm duration, etc (Hall and Sobel, 2013).

Two classes of methods are widely used to estimate the probability of occurrence of extreme sea levels: direct and indirect methods. Indirect methods model separately the deterministic and the stochastic components of $z(t)$ then recombined by convolution. Examples are the joint probability method (Pugh and Vassie, 1979, 1980), the revised joint probability method (Tawn and Vassie, 1989), the exceedance probability method (Middleton and Thompson, 1986; Hamon and Middleton, 1989), and the empirical simulation technique (Scheffner et al., 1996; Goring et al., 2011). Direct methods, such as the one adopted here, analyze observed values compounding the astronomical and stochastic storm-surge component. Direct methods mostly differ based on the analysis approach adopted, such as the annual maxima method (Jenkinson, 1955; Gumbel, 1958), the peaks-over-threshold method (Davison and Smith, 1990), or the r-largest method (Smith, 1986; Tawn, 1988). Here, we study the distribution of the sum, $h(t)$, of the contributions from the deterministic tide and the stochastic surge:

$$h(t) = z(t) - msl(t) \tag{2}$$

From a statistical point of view, this choice is justified by the fact that the random arrival of storms adds a stochastic surge contribution at unpredictable times, thereby causing $h(t)$ to be values from a random variable, even though it contains a deterministic component. The presence of a deterministic component of course does imply a strong auto-correlation in the observed signal, which will be subsequently filtered out by suitable signal processing described below.

$msl(t)$, is here computed as the yearly average of daily levels. The yearly average is chosen, rather than the customary 19-year average that eliminates all tidal periodicities, however small in amplitude, to better capture the surge contribution that causes the water level to deviate during a storm with respect to the "current", yearly, value of $msl(t)$. Once $h(t)$ is computed by removing $msl(t)$ from recorded levels, all local maxima of $h(t)$, or water level peaks, are identified and their values constitute the basis for subsequent analyses of (i) long-term trends study of maximum yearly departures from the average mean sea level (two-tail Mann-Kendall test, Mann (1945)), and (ii) statistical inference of past coastal flooding events and their potential future changes.

In the following discussion, we will use the terms "total water level" and "coastal water level" when referring to the quantities $z(t)$ and $h(t)$ respectively.

### 2.2.2 Extreme Value Theory

As highlighted by Serinaldi and Kilsby (2014), the EVT deals with the asymptotic distributional behaviour of two types of data modeled with two well-known approaches, namely the so-called Block Maxima (BM) and Peaks-Over-Threshold (POT). The first type models the maximum values extracted from blocks of fixed length, whereas the second one models all the exceedances of high threshold. The cornerstones of the EVT are two theorems: the Fisher-Tippett-Gnedenko theorem (also known as the three types theorem, Fisher and Tippett (1928); Gnedenko (1943); Gumbel (1958)) and the Pickands-Balkema-de Haan theorem (also known as the second theorem of EVT, Balkema and de Haan (1974); Pickands (1975)).

According to the three type theorem, there are three possible non-degenerate distribution functions which can arise as limiting distributions of extremes of random sample: (i) the Gumbel distribution or type I, (ii) the Fréchet distribution or type II, and (iii) the reverse-Weibull distribution or type III. The above three limiting distribution laws can be combined into a single family of three-parameter distribution known as the Generalized Extreme Value (GEV) distribution given by:

$$G(x; \mu, \psi, \xi) = exp\{-[1 + \frac{\xi}{\psi} \cdot (x - \mu)]\}^{-1/\xi} \tag{3}$$

defined on the region for which $\{x : 1 + \frac{\xi}{\psi} \cdot (x - \mu) > 0\}$. In Eq. 3 $\mu \in (-\infty, +\infty)$ is a location parameter, $\psi > 0$ is a scale parameter, and $\xi \in (-\infty, +\infty)$ is a shape parameter which controls the nature of the tail distribution (Frechét type for $\xi > 0$, Gumbel type for $\xi = 0$, and reverse-Weibull type for $\xi < 0$).

The second theorem of EVT defines a method to model the tail of distribution above a threshold value (Davison and Smith, 1990). In particular, the theorem states that for a large enough threshold value, $u$, the distribution of exceedances of some high threshold ($y = X - u$, where $X$ is a sequence of i.i.d. random variables), is described by a Generalized Pareto Distribution

(GPD), which has the following cumulative distribution function:

$$G(y; \sigma_u, \xi) = 1 - (1 + \frac{\xi}{\sigma_u} \cdot y)^{-1/\xi} \tag{4}$$

defined on $\{y : y > 0 \text{ and } (1 + \frac{\xi}{\sigma_u} \cdot y > 0)\}$, where $\sigma_u$ and $\xi$ are the scale and shape parameters respectively.

There is a link between these two distributions according to which if block maxima have approximate distribution GEV, then threshold excesses have corresponding approximate distribution within the generalized Pareto family and vice versa, GEV can be obtained from GPD under two appropriate conditions (i.e., the occurrences are Poisson-distributed and excesses over threshold come from a GPD). The duality between Eq. 3 and Eq. 4 means that the GPD parameters of the excesses are

uniquely determined by those of the associated GEV distribution of block maxima (see e.g., Coles (2001)). In particular, the shape parameter, $\xi$, is equal to that of the corresponding GEV distribution and the scale parameters of the two distributions are related by $\sigma_u = \sigma + \xi \cdot (u - y)$.

The interested reader can refers to Coles (2001) for a detailed description of statistical methods for extremes in hydrology or Papalexiou and Koutsoyiannis (2013) for a recent overview of the history of the EVT.

### 2.2.3    The Metastatistical Extreme Value Distribution

The typical EVT derivation starts from the premise that the maximum value among $n$ realizations of a random variable ($M_n$) is distributed according to the cumulative distribution function $P(M_n \leq x) = G(x) = F(x; \overrightarrow{\theta})^n$ (where, as customary, a capital letter indicates the random variable and a small cap letter indicates a value of the random variable). This approach assumes that the $n$ values of the random variable of interest are generated by the same distribution, the "ordinary value" distribution

$F(x; \overrightarrow{\theta})$, and are thus independent and identically distributed. $n$ is the number of events in a block, such that $G(x)$ is the cumulative distribution of the block maxima. The classical EVT assumes that either the number of events/block is large (asymptotic hypothesis, leading to the GEV-BM formulation) or that the number of events/block above a high threshold is distributed according to a Poisson distribution (POT-GPD formulation). The recently-proposed Metastatistical Extreme Value Distribution (Marani and Ignaccolo, 2015) is a doubly stochastic approach (Dubey, 1968; Beck and Cohen, 2003) that relaxes these hy-

potheses by considering both the parameters ($\overrightarrow{\theta}$) of the ordinary value probability distribution and the number of events/block to be random variables. Hence, the MEVD cumulative distribution of block maxima (estimated using a much greater sample than just yearly maxima used in the BM approach) is then defined as the compound probability:

$$G(x) = \sum_{n=1}^{+\infty} \int_{\Omega_{\overrightarrow{\Theta}}} F(x; \overrightarrow{\theta})^n g(n, \overrightarrow{\theta}) d\overrightarrow{\theta} \tag{5}$$

where $g(n, \overrightarrow{\theta})$ is the joint probability distribution of the number of events in a block and of the parameters vector (discrete in $N$ and continuous in $\overrightarrow{\Theta}$), $\Omega_{\overrightarrow{\Theta}}$ is the population of all possible parameter values.

For practical applications, the MEVD can be approximated by substituting ensemble average in Eq. 5 with the sample average

computed over all the blocks in the time series, obtaining:

$$G(x) \cong \frac{1}{M} \sum_{j=1}^{M} F(x; \theta_j)^{n_j} \tag{6}$$

where $M$ is the number of blocks in the historical record, $F(x; \overrightarrow{\theta_j})$ is the cumulative distribution of ordinary values in the $j^{th}$ block, and $n_j$ is the number of events in the $j^{th}$ block. A common choice for the block length is 1 year. Note that the values of the parameters $\overrightarrow{\theta_j}$ may be estimated on Estimation Windows (EW) with length that is different from block length. For example, if the block length is 1 year, it may be advantageous to estimate parameter values on longer time slices to ensure, depending on the rate of event occurrence, that a reliable estimation of the parameters is possible. Miniussi and Marani (2020) in applications to daily rainfall extremes find that, when the number of events per year is less than 20-25 events/years, then the optimal EW length may be greater than one year.

It is interesting to note that the POT approach, briefly described above, can be thought of as a particular case of MEVD. In fact, Zorzetto et al. (2016) highlight that if one assumes (i) $x$ to be the excess over a high threshold, (ii) $F(x; \overrightarrow{\theta_j})$ to be a Generalized Pareto Distribution (with fixed, deterministic parameters), and (iii) $n$ to be generated by a Poisson distribution, then the GEV distribution is recovered as a particular case of the MEVD by means of the POT approach.

MEVD has been applied in several earth-science contexts. In rainfall extremes estimates, the ordinary value distribution is assumed to be Weibull when applied to point daily rainfall (Marani and Ignaccolo, 2015; Zorzetto et al., 2016; Schellander et al., 2019; Miniussi and Marani, 2020; Miniussi et al., 2020b), point sub-daily rainfall (Marra et al., 2018), and satellite rainfall estimates (Zorzetto and Marani, 2019, 2020). For flood across the United States, Miniussi et al. (2020a) propose to adopt a Gamma distribution for $F(x; \overrightarrow{\theta_j})$. Hosseini et al. (2020) describe Atlantic hurricane intensities using a Generalized Pareto ordinary value distribution. In all cases the appropriate form for the underlying ordinary value distribution was identified by minimizing the estimation uncertainty within a cross-validation approach, which is also followed here. In this particular application to extreme coastal water levels, three candidate probability distributions for $F(x; \overrightarrow{\theta_j})$ in Eq. 6 are tested, i.e. the Gamma, Weibull and Generalized Pareto distributions. Based on the comparative evaluation of the performance of these distributions, e.g. using diagnostic quantile-quantile scatter plots, the Generalized Pareto distribution emerged as the best model for the "ordinary" coastal water level values.

In the present context, we define as ordinary values any coastal water elevation (i.e. the maximum water level reached during a storm event) greater than a site-specific threshold value. This threshold is chosen to be as small as possible (differently from the POT approach), to retain as much of the observational information as possible, and will be dependent on the magnitude of the local tidal range (sea level difference between high and low water level over a tidal cycle) and of storm contributions. Additionally, the threshold is set to be large enough to filter out coastal water level peaks that are likely fully determined by tidal fluctuation, in the absence of any storm contribution. Given the above constraints, we also choose the threshold value that minimizes the estimation error under the MEVD framework.

As suggested by several rainfall applications, ordinary distribution parameters are here estimated using the Probability Weighted Moments (PWMs) method in non-overlapping estimation windows of 5 years. In the present application, the op-

timal estimation window length was set to 5 years to obtain a more robust parameters estimation, especially when few values in each year are available. PWM estimation, introduced by Greenwood et al. (1979), is widely applied because of its good
performance, particularly in the presence of small sample sizes, its reduced estimation bias and sensitivity to the presence of outliers in the data (Hosking et al., 1985; Hosking and Wallis, 1987; Hosking, 1990).

### 2.2.4 Selection of independent events

The GEV-based approaches are fit on either annual peak maxima (GEV-BM) or on a few water level peaks over a high threshold (POT-GPD), which can be assumed to be realizations of independent stochastic variables. The MEVD requires that all ordinary
values (coastal water level peaks in this case) within one block may be assumed to be realizations from independent random variables. This hypothesis, in turn, requires that observed peaks are filtered to only retain events that may be considered to be independent, through a de-clustering process (Coles, 2001; Ferro and Segers, 2003; Beirlant et al., 2004; Bommier, 2014; Marra et al., 2018). Several criteria have been developed for such processing of the data. A common criterion sets the minimal time separation, or lag ($\tau$), for two events to be considered independent. Intuitively, high water-level events separated by a
sufficiently long time period are reasonably caused by distinct storm events. However, when analyzing the water level with respect to current mean sea level, a quantity that contains the deterministic tidal contribution, dependence may be expected to be present also for large lags. In theory, some dependence is present for lags up to the longest periodicity in the tidal signal (18.61 years). In practice, as the dependence in the tidal signal decreases for increasing lag, one expects that a much shorter threshold time lag will be sufficient to make sure that only independent events are considered. The analysis of the correlograms
of selected coastal water level peaks shows that some correlation persists also for long time lags and also in the de-clustered time series. Even though the strength of this correlation is relatively small (the autocorrelation function, ACF, is always less than 0.3), it could impact the ability of the MEVD, which assumes independence, to capture observed extreme behaviour. The de-clustering process does significantly decrease correlation, as may be seen by comparing Figure S1 (ACF prior to de-clustering) and Figure S2 (after de-clustering). Interestingly, it is seen that the tidal contribution (that generates periodicities in the ACF)
is strongly visible in Venice and Newlyn, while it is quite small in Hornbæk and Marseille. The underlying tidally-induced correlation becomes more clearly visible after de-clustering also in Hornbæk and Marseille. We note that the existing literature implementing de-clustering approaches to coastal level signals normally focuses on studying the storm-surge component only. As result, it uses threshold time lag values that are smaller than those adopted here because characteristic correlation times of the surge component are significantly smaller than those associated with the sum given by the combination of surge and
tidal components. For example, the independence between two consecutive storm surge events in southern Europe has been found to be achieved with a threshold lag of 3 days (Cid et al., 2015). A threshold separation of one day between consecutive events is imposed by Tebaldi et al. (2012) in their analysis of storm surges along the US coast. Haigh et al. (2010) adopt a threshold lag of 30 h in the English Channel, while Bernardara et al. (2011) assume a 72 h independence criterion. After exploring values between 24 h and several days, we adopt a threshold lag of 30 days, which yielded the minimum estimation
error under the MEVD approach, and is consistent with the main lunar periodicity. The result of this de-clustering process is a

set of independent events with magnitudes $h_k$, whose number $n_j$ in year - or block - $j$ is a realization of a random variable as illustrated in Eq.s 5 and 6.

### 2.2.5 Cross-validation procedure

Statistical modelling aims to use sample information to infer the probability distribution of the population from which the data are extracted. This inference is uncertain due to imperfect parameter estimates and to the possible inability of the chosen distribution to capture the statistical properties of the underlying population. Although these sources of uncertainty are inherent in any statistical model, their impact can be minimized by a careful choice of the model and by an effective use of all sources of information (Coles, 2001). In many applications uncertainty is estimated by means of goodness-of-fit measures, which quantify how well the distribution compares to the sample on which it was fitted. However, this procedure does not provide a measure of the predictive uncertainty encountered when trying to estimate the probability of occurrence of the "next", yet unobserved value. In this study, we evaluate the performance in high-quantile estimation associated with the use of the MEVD and the GEV distribution, by means of a cross-validation (CV) procedure, in which model predictions of the probability of occurrence are compared to frequencies from data that were not used in the estimation of model parameters. This is possible by dividing observations into two sets of independent data: the estimation set is the sample from which model parameters are estimated and the test set is the sample with which model predictions are compared.

The procedure can be summarized as follows: a) we randomly reshuffle the observational years on record while keeping all the water level independent peaks in their original year to 1) preserve both the ordinary value frequency distribution in each year and the distribution of the number of events/year, and 2) remove possible non-stationarity and correlation in the time series; b) we divide the observational sample into two independent sub-samples obtained by randomly selecting $S$ years from the original time series of length $M$: this sub-sample (in the following "calibration sample") is used for parameter estimation, while data in the remaining $V = M - S$ years are used for testing (in the following "validation - or test - sample"); c) as usual in frequency analysis, we associate to each observed yearly maximum, $x_i$, an empirical frequency value given by Weibull's estimator $F_i = i/(V + 1)$ where $i$ is the rank of $x_i$ in the list of yearly maxima sorted in ascending order, and $V = M - S$ is the sample size in the validation sub-sample. The return period $Tr$ associated with each yearly maximum is then simply $Tr_i = 1/(1 - F_i)$; d) we estimate the GEV and MEVD quantiles using the parameter values estimated in step b) from the calibration sub-sample; e) focusing on the validation sub-sample, in every realization (for $p = 1, \ldots, Nr$; $Nr = 1000$ here) and for a fixed mean recurrence time ($Tr$), we compute the Non Dimensional Error between the estimated and observed quantiles as follows: $NDE_p(S, Tr) = [h_{(est,p)}(S, Tr) - h_{(obs,p)}(S, Tr)]/h_{(obs,p)}(S, Tr)$; f) we repeat the CV scheme above $Nr$ times. This procedure is performed for different calibration sample sizes ($S$ = 5, 10, 20, and 30 years) to evaluate how estimation uncertainty varies with return period and calibration sample size.

### 2.2.6 Future total water level projections

Future increases in the frequency of extreme total water levels (i.e. the variable previously referred as $z(t)$) due to climate change will have serious impacts on coastal regions. These impacts will vary temporally and regionally, depending on (i)

the local relative mean sea-level rise (including possible subsidence or uplift), (ii) current storm-surge intensity probability

distributions, and (iii) changes in the dominant meteorological dynamics. In this particular application to extreme coastal water

levels (i.e. the sum given by the combination of the water level setup, induced by meteorological forcing, and the astronomical

tide), only the first two factors are considered.

It is very likely that sea-level rise will continue to accelerate over time, thereby increasing the frequency of extreme sea level

events, leading to severe flooding in many low-lying coastal cities and small islands (Oppenheimer et al., 2019). Various

techniques have been used to study possible changes in coastal flooding hazard (e.g., McInnes et al., 2013; Vousdoukas et al.,

2016). Several authors have found that past variations in the frequency of occurrence of extreme sea levels have been primarily

determined by changes in mean sea level (e.g., Zhang et al., 2000; Woodworth and Blackman, 2004; Lowe et al., 2010;

Menéndez and Woodworth, 2010; Haigh et al., 2014b; Wahl et al., 2017). This implies that effects of variations in storminess

(e.g., magnitude, trajectories and frequency) have been small in the observational record, compared to the dominant effects

of mean sea-level changes (Haigh et al., 2014a). This notion is also confirmed by our trend analyses of maximum yearly

departures from the average sea level (see §3.1), which fail to detect trends in the maximum difference between total sea level

and concurrent mean sea level except at one of the sites (Venice), where it is smaller (0.7 mm/yr) than past and projected rates

of sea-level rise (respectively $\sim$3.0 mm/yr and $\sim$8.0 mm/yr by the end of the century, according to the RCP8.5 IPCC scenario).

Based on these elements, here we estimate the probability of future total water levels along European coastlines by assuming

that changes in the tidal and storm-surge components are negligible with respect to changes in mean sea-level, an assumption

common to previous approaches (Araújo and Pugh, 2008; Haigh et al., 2010; Tebaldi et al., 2012).

To assess how the exceedance probabilities of extreme total water levels might change in the future, the projections of sea-level

rise through 2100 from the IPCC's Fifth Assessment Report (AR5) are used. In particular, we explore an intermediate (RCP4.5)

and an extreme scenario (RCP8.5), using CMIP5 model outputs from the "Integrated Climate Data Center" (ICDC) database

(University of Hamburg: https://icdc.cen.uni-hamburg.de/en/ar5-slr.html).

For each tide gauge, our approach can be summarized as follows: 1) we infer the probability distribution of extreme coastal

water levels (annual maxima) from observed independent events whose intensity (maximum coastal water level attained, $h_k$)

is defined with respect to the concurrent mean sea level computed on a yearly basis; 2) we estimate the future probability of

extreme total water levels by translating extreme level quantile estimates upward according to location-specific projections of

310 mean sea level in the year 2100 (thereby implicitly assuming subsidence/uplift to be negligible).

### 2.2.7 Return period

One of the main objectives of frequency analysis is to calculate the average recurrence interval or return period. It is a widely

used concept in hydrological and geophysical risk analysis. If a process is stationary, the return period ($Tr$) of an event

magnitude is defined as the average time elapsing between two consecutive exceedances of this magnitude. Alternatively, it

may be said that a magnitude value is expected to be exceeded, on average, in each return period. If the yearly-maximum

magnitude $h$ is exceeded on average once in $Tr$-years, then its exceedance probability, $E(h) = 1 - G(h)$, in a given year is:

$$E(h) = P[H \geq h] = \frac{1}{Tr(h)}$$

Therefore, the return period of the level value $h$ is the inverse of the probability of exceedance and can be expressed as a function of the cumulative distribution, $G(h)$, of annual maxima, e.g. through the MEVD (Eq. 6):

$$Tr(h) = \frac{1}{E(h)} = \frac{1}{1 - G(h)} \qquad (7)$$

Because for a fixed value of mean sea level there is a one-to-one relation between the value of the sum of the astronomical and the storm surge contribution, $h$, and the total water level, $z = h + msl$, one can write $G_h(h) = P[H > h] = P[H > z - msl] = P[Z - msl > z - msl] = P[Z > z] = G_z(h)$, such that Eq. 7 can be used, once the cumulative distribution is known and for each (time-dependent) value of $msl$, to determine the return period of the total water level (at the time when $msl$ is evaluated):

$$Tr(z) = \frac{1}{1 - G_z(h)} = \frac{1}{1 - G_h(h)} = \frac{1}{1 - G(z - msl)} \qquad (8)$$

Based on the hypothesis introduced in §2.2.6 that mean sea-level rise is the dominant effect in future coastal flooding, we assume that the characteristics of the extremes (i.e. the parameters of the GPDs defining the MEVD) remain valid in future scenarios. Eq. 8 clarifies that the return period of a fixed value $z$ decreases as $msl$ increases, basically because for higher values of $msl$ a smaller value of $h$ is needed to achieve the same total water level $z$. This decrease is non-linear, due to the nonlinear form of the right-hand side in Eq. 8.

## 3 Results and discussion

### 3.1 Mann-Kendall trend analysis

We start by computing mean sea level on yearly basis and by subtracting it from observed total water level. The first question that we want explore is the presence of log-term trends, unrelated to sea-level rise and associated to other factors (e.g., human-induced factors, morphological variations, etc.), in the "cleaned up" signal, i.e. the observed measurements without mean sea level. To answer this question, in this work we focus on the deviation of yearly maxima from yearly mean sea level and test for the presence of trend by the two-tail Mann-Kendall test (Mann, 1945). Figure 2 summarizes results for each location explored. From a first visual inspection of Figure 2, the Venice (1872-2019) and Hornbæk (1891-2012) time series appear to show an increasing trend in the deviations of yearly maxima from yearly mean sea level (blue line) of different magnitudes. On the contrary, Marseille sea level observations (1985-2018) seem to be characterized by a decreasing trend. Finally, the Newlyn historical record (1915-2016) displays a fairly constant signal with no noticeable variations. The application of the Mann-Kendall test reveals a partly different story. The test rejects the hypothesis of the absence of trend at the 95% confidence level, only for the Venice site (p-value$^{\text{Venice}}$ = 0.014). This result suggests that the increase of the yearly maximum deviations from yearly mean sea level may be a direct results of the local morphological variations of lagoon channels where the tidal wave

propagates (whereby dissipation of the wave is reduced), and/or land subsidence. On the contrary, at the remaining locations, the null hypothesis of no trend cannot be rejected (p-value[Hornbæk] = 0.352, p-value[Marseille] = 0.110, and p-value[Newlyn] = 0.997).

The results obtained from these analyses support the validity of the hypothesis that mean sea-level rise is the dominant factor in determining the future frequency of coastal flooding (see §2.2.6). For the tests performed here to compare different extreme-value statistical models, the possible presence of trends (e.g., in Venice) is irrelevant, since such tests are performed by first reshuffling observed values, thereby eliminating any existing trend, albeit small.

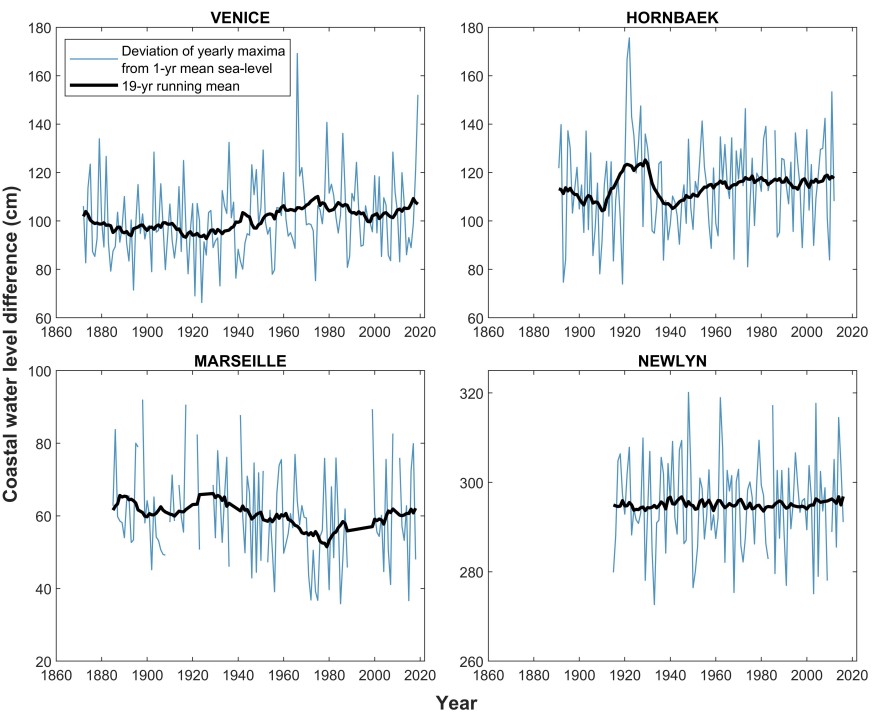

**Figure 2.** Deviation of yearly maxima from yearly mean sea level (blue line) and 19-yr running mean (black line) calculated for Venice (IT), Hornbæk (DK), Marseille (FR), and Newlyn (UK).

## 3.2    Extreme value analysis

The MEVD formulation requires the choice of an optimal distribution of ordinary values that can represent the characteristics of the natural phenomenon under analysis. Different candidate distributions for the $F(x; \overrightarrow{\theta_j})$ in Eq. 6 are evaluated and the most suitable distribution is selected on the basis of the *CV* procedure comparing the MEVD-estimated quantiles with the observed ones. As previously introduced in §2.2.3, according to different tests, the appropriate distribution to model the ordinary sea-level values is the Generalized Pareto Distribution (GPD). We highlight that the GPD used in the MEVD framework is

obtained by imposing a small threshold (differently from the high threshold adopted in the POT-GPD approach) to capture the distribution of the main body of the probability distribution of the ordinary events and does not require the event arrival process

to be Poisson (Marani and Zorzetto, 2019).

As mentioned above (§2.2.4), the independence between two consecutive coastal water level events is guaranteed by imposing a minimum time lag. Firstly, we select daily maxima sea levels from the original record; secondly, define as independent events those that are separated by at least 30 days. Subsequently, the samples used for statistical inference are built as follows: 1) GEV-BM: the yearly maxima are selected; 2) POT-GPD: as proposed by Coles (2001), the optimal threshold ($u$) is determined by studying the stability of the GPD shape ($\xi$) and modified scale ($\sigma^* = \sigma_u - \xi u$) parameters estimated using a wide range of values of $u$. Using this method, the following threshold values were identified: 65 cm (Venice), 50 cm (Hornbæk), 35 cm (Marseille) and 260 cm (Newlyn); 3) MEVD: all the independent coastal water level events above a low threshold are used to fit the probability distributions of ordinary values. The optimal threshold to apply to all the independent events for extrapolating the ordinary values sample, is chosen by testing different threshold values and evaluating the goodness-of-fit of the distribution using diagnostic graphical plots. According to the selection criteria described in §2.2.3, the low thresholds adopted in the four study sites are 59 cm for Venice, 40 cm for Hornbæk, 25 cm for Marseille, and 250 cm for Newlyn. For every observed site, Table 2 and Figure S3 display the gradual increase in the number of independent events (i.e. annual maxima, exceedances over the threshold, and ordinary values) used to infer the distributions when moving from GEV-BM, POT-GPD to MEVD approaches.

Considering the above threshold values, the observed and estimated distributions of coastal water level are compared by plotting their quantiles against each other. By comparing measures of in-sample and out-of sample test predictive accuracy, the results are presented by means of quantile-quantile (QQ) plots. The reader can refer to Figure 3 (or supplementary Figures S4, S5, S6, S7 and S8) to compare the results obtained with the MEVD framework (or the GEV-based approaches - GEV-BM and POT-GPD - vs. the MEVD formulation) for the four sites analyzed. QQ-plots are obtained as a result of the CV procedure with 1,000 random realizations and sample size: a) $S = 30$ years (in-sample-test on the left column); b) $V = M - S$ years (out-of-sample test on the right column). The colours represent the density of points around the 45° line (i.e. the line of equality). This highlights how the estimated quantiles are closely comparable with the observed ones for all the three approaches tested and for both the sample size explored ($S$ and $V$). In particular, if the reader looks at the supplementary Figures from S4 to S8 and if out-of-sample performance is considered, it is difficult to quantify which distribution is the best due to a large variability in the estimates. Overall, if only the MEVD performance is investigated, the reader can look to the right column (out-of-sample test) in Figure 3, where the results display that the MEVD formulation performs similarly for all sites analyzed. In particular, it proves to be a good model for lower/intermediate quantiles but shows variability in the estimates for higher quantiles.

We now focus on evaluating the performance of the three approaches (GEV-BM, POT-GPD and MEVD) in high-quantile estimation. We explore the predictive performance of the MEVD and GEV distribution as a function of the NDE (§2.2.5) computed for the maximum return period, $Tr_{max} = M - S + 1$, associated with the maximum value in each test sub-samples that we randomly extract in the CV approach. The use of NDE metric allows to easily characterize and compare models estimation uncertainty associated with fixed return time of interest and the variation of the calibration sample size (from 5 to 30 years). The results are summarized by means of box-plots (Figure 4) and kernel density estimates computed for a calibration sample size of 30 years (Figure 5). Table 3 summarizes the main results underlying the chosen evaluation metric. When we

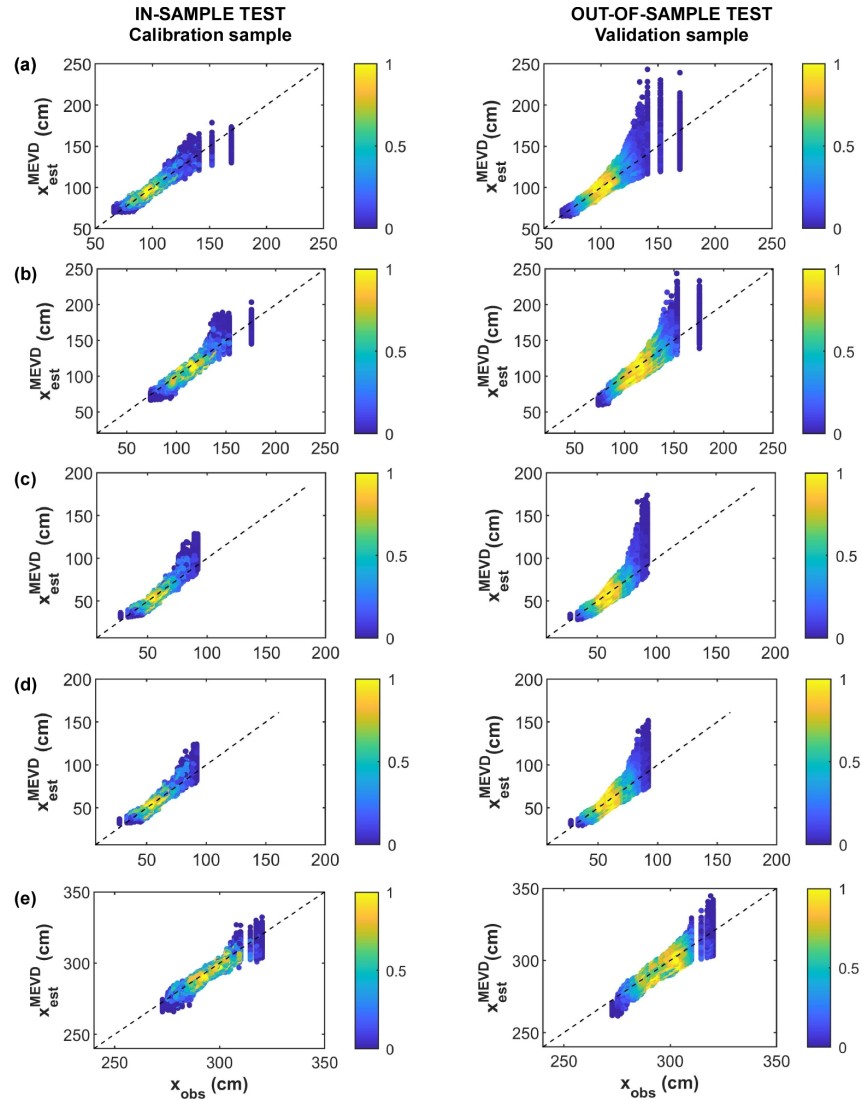

**Figure 3.** QQ-plots of extreme coastal water level quantiles, computed with the MEVD framework, for the (a) Venice (IT), (b) Hornbæk (DK), (c-d) Marseille (FR), and (e) Newlyn (UK) sites. The MEVD parameters estimations are based on non-overlapping sub-samples of fixed size (5 years), while subplots indicated with the letter (d) display the QQ-plots obtained with MEVD parameters estimations based on data from the whole calibration sample size. The plots are obtained as a result of the cross validation method used to test the global performance of the models and are estimated for 1,000 random realizations and for sample size: a) S = 30 years (in-sample-test on the left column); b) V = M-S years (out-of-sample test on the right column). The colours represent the points density around the 45° line (black dashed line) corresponding to the best fit.

**Table 2.** Total number of independent events and average number of events/year for all the gauge stations explored.

| Site name | | Independent events | | |
|---|---|---|---|---|
| | | BM | POT | MEVD |
| Venice | Total | 148 | 605 | 775 |
| | N. events/year | 1 | 4.08 | 5.23 |
| Hornbæk | Total | 121 | 595 | 736 |
| | N. events/year | 1 | 4.91 | 6.08 |
| Marseille | Total | 106 | 275 | 489 |
| | N. events/year | 1 | 2.57 | 4.61 |
| Newlyn | Total | 100 | 399 | 520 |
| | N. events/year | 1 | 3.99 | 5.20 |

focus on the case of a short sample (5 years), different sites display variable results: I) the GEV and MEVD approaches perform similarly for Venice (Figure 4(a)) and Hornbæk (Figure 4(b)) with similar interquartile ranges and underestimations of the actual quantile; II) for the Newlyn gauge station (Figure 4(d)) the GEV-BM distribution yields better results, even though the POT and MEVD median error are also close to zero. On the contrary, when we consider longer calibration sample sizes (from 10 to 30 years), the MEVD-based estimates outperform the traditional approaches for most gauge stations explored: I) results for the Venice site confirm the robustness of the MEVD with respect to the GEV distribution especially for calibration sample size equal to 30 years. In this case, the median error in the MEVD estimates tends to be closer to zero (-0.004), corresponding to approximately unbiased estimates; II) the Hornbæk station displays similar results to those for Venice and the MEVD-based estimates become more reliable when we consider a calibration sample size greater than 10-20 years; III) Newlyn estimation errors show a trade-off between the BM method and MEVD for calibration sample size equal to 20 and 30 years.

Results for the Marseille site show a peculiar behaviour that requires a specific discussion. In this case, the application of the traditional extreme value theory is advantageous when compared with the MEVD (Figure 4(c)). In order to better understand the application to the Marseille site, we performed MEVD parameter estimation using two approaches: 1) estimation based on non-overlapping calibration samples of fixed size (5 years as for the other sites); 2) parameter estimation on data from the whole calibration sample. The comparison of the results from these two set-ups confirms that when longer time slices are used for estimating GPD parameters (black colour in Figure 4(c)), the MEVD performance is improved (for example when we consider $S = 30$ years, MEVD median$_{[S\text{-year}_{window}]} = 0.17$ vs. MEVD median$_{[5\text{-year}_{window}]} = 0.35$), though it does not yet match the results obtained from GEV-BM approach (GEV-BM median error = 0.016). This can be explained by considering sea-level peaks occur in Marseille about once every year on average. In this case GEV-BM is advantageous because the small number of events/year does not provide a more numerous calibration sample with respect to the sample of annual maxima. This result confirms the conclusion by Miniussi and Marani (2020), according to which the selection of the estimation window size for fitting the ordinary value distribution strongly depends on the average number of extreme events/year.

We also provide a comparative analysis between the three methods to evaluate if the tested extreme value distributions are representative of the entire range of return times of interest. To achieve this purpose, we evaluate methods performance also

**Table 3.**

Results of the evaluation metric obtained for all the gauge stations and for calibration sample sizes ($S$) equal to 5 and 30 years. In the case of the Marseille site, text in bold refers to MEVD parameter estimation based on data from the whole calibration sample size.

| Site name | Variables | S = 5 yrs | | | S = 30 yrs | | |
|---|---|---|---|---|---|---|---|
| | | BM | POT | MEVD | BM | POT | MEVD |
| Venice | NDE-median | -0.160 | -0.175 | -0.178 | -0.133 | -0.158 | -0.004 |
| | NDE-mean | -0.069 | -0.101 | -0.116 | -0.087 | -0.133 | 0.024 |
| | NDE-std | 0.366 | 0.274 | 0.267 | 0.156 | 0.113 | 0.155 |
| Hornbæk | NDE-median | -0.119 | -0.104 | -0.113 | -0.113 | -0.115 | 0.056 |
| | NDE-mean | -0.069 | -0.101 | -0.116 | -0.068 | -0.087 | 0.077 |
| | NDE-std | 0.366 | 0.274 | 0.267 | 0.113 | 0.100 | 0.131 |
| Marseille | NDE-median | -0.0003 | 0.059 | 0.172 | 0.016 | 0.047 | 0.357 **0.172** |
| | NDE-mean | 0.045 | 0.129 | 0.262 | 0.013 | 0.050 | 0.374 **0.183** |
| | NDE-std | 0.252 | 0.350 | 0.421 | 0.072 | 0.115 | 0.178 **0.140** |
| Newlyn | NDE-median | -0.010 | -0.030 | -0.033 | -0.003 | -0.032 | 0.0008 |
| | NDE-mean | 0.003 | -0.022 | -0.026 | -0.002 | -0.031 | 0.002 |
| | NDE-std | 0.050 | 0.042 | 0.042 | 0.016 | 0.014 | 0.021 |

for intermediate $Tr$ values, greater than the calibration sample size, since for $Tr < S$ the empirical quantiles can be used. We perform this additional analysis for the Venice, Hornbæk and Newlyn sites. Figure 6 summarizes the results obtained by estimating the probability distribution parameters on 30-year calibration sub-samples. The analyses suggest that when we focus on the median error associated with moderate values of the return period, GEV-BM displays an overall greater robustness (e.g., in the case of Venice and Hornbæk sites) with respect to POT-GPD and MEVD, which exhibit greater fluctuations. In particular, results show that MEVD is a good model for the highest values of the return period, but exhibit a greater absolute value of the estimation error for smaller $Tr$. Overall, the results suggest that no single approach is clearly superior at all values of $Tr$, due to a large variability in the estimates. For example, for the Venice site there is a decrease (in many cases an unbiased estimates) in the MEVD-NDE values for intermediate $Tr$ (between 85 and 105 years) while for greater $Tr$ values (but smaller than $Tr_{max}$) the error shows an overestimation of the actual quantile with respect to traditional approaches (which exhibit an underestimation tendency). To be more specific, if $Tr > 105$ years are considered, MEVD yields error estimates between zero and <10%, while errors associated with GEV-BM and POT-GPD lie between zero and <-20%. The Hornbæk site shows similar results to the Venice site, while Newlyn's results exhibit more fluctuations for large $Tr$ values with much reduced smaller amplitudes and values of the NDE.

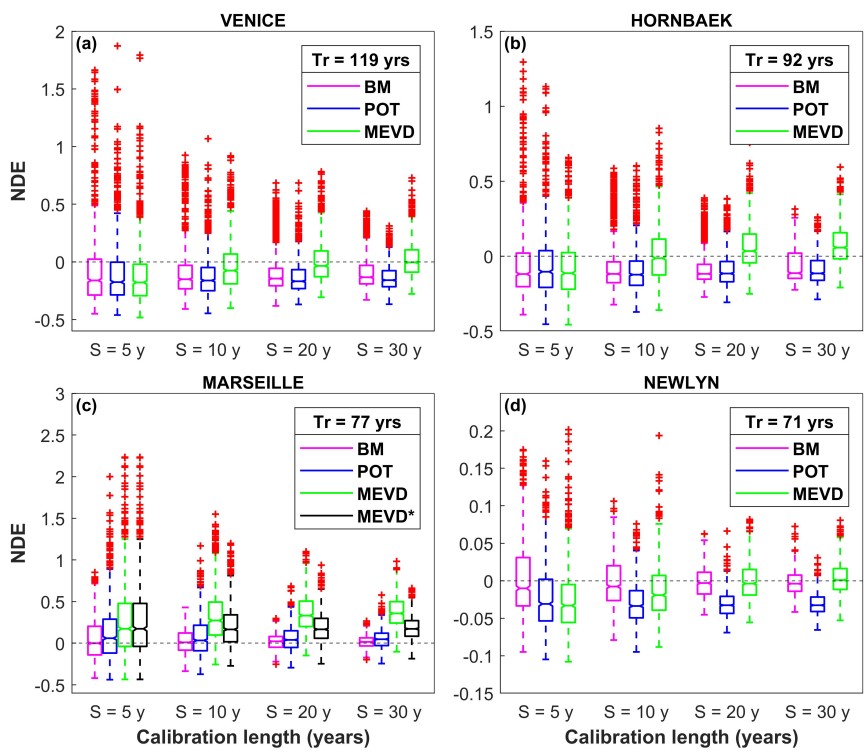

**Figure 4.** Distribution of the Non Dimensional Error (NDE) for maximum sample return period ($Tr$) represented by means of box-plots at given gauge stations explored: (a) Venice (IT), (b) Hornbæk (DK), (c) Marseille (FR), (d) Newlyn. In the case of the Marseille (FR) site, MEVD parameter estimation is based: 1) green colour: on non-overlapping sub-samples of fixed size (5 years), and 2) black colour: on data from the whole calibration sample.

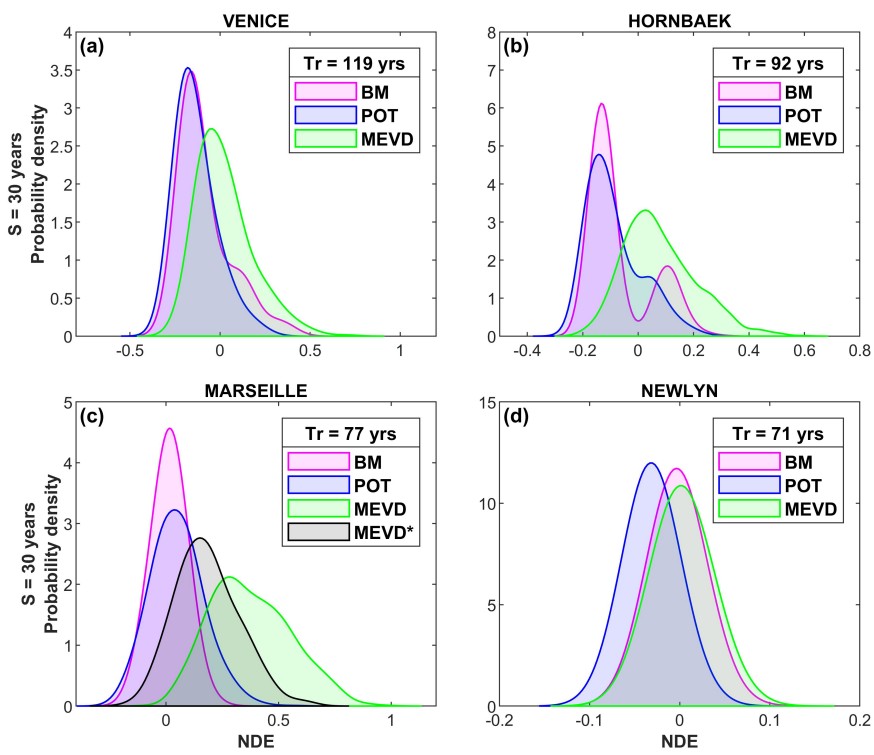

**Figure 5.** Kernel density estimates for the Non Dimensional Error (NDE) distributions obtained with a calibration sample size ($S$) of 30 years and maximum return period ($Tr$) at given gauge stations explored: (a) Venice (IT), (b) Hornbæk (DK), (c) Marseille (FR), (d) Newlyn (UK). In the case of the Marseille (FR) site, MEVD parameter estimation is based: 1) green colour: on non-overlapping sub-samples of fixed size (5 years), and 2) black colour: on data from the whole calibration sample.

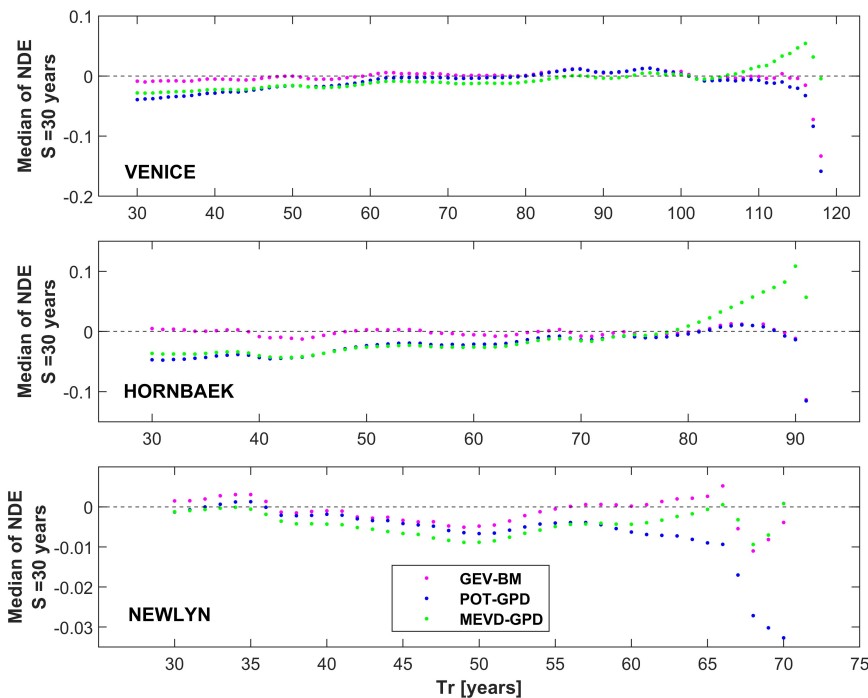

**Figure 6.** Median of the non-dimensional error (NDE) for return period greater than the calibration sample size in test sub-sample for the GEV-BM, POT-GPD and MEVD approaches (magenta, blue and green dots respectively). The results are obtained for the Venice (IT), Hornbæk (DK) and Newlyn (UK) sites and by estimating the distribution parameters on 30-year calibration sub-samples.

## 3.3 Future projections of extreme total water levels

We next explore how sea-level rise may influence the frequency of extreme total water levels across the sites analyzed. As described in §2.2.6, we only evaluate the influence of an increased mean sea level, i.e. we do not address possible changes in
storm regimes (e.g., see Tebaldi et al., 2012).

We use site-specific sea-level projections from IPCC's AR5 (Church et al., 2013), which indicate an accelerating sea-level rise in all four observation sites (for each gauge station under analysis, the reader can refer to the panels (a), (c), (e), and (g) in Figure 7), with expected water level increases by the end of the century (RCP8.5) of 48 cm in Venice, 52 cm in Hornbæk, 59 cm in Newlyn and 54 cm in Marseille. The panels (b), (d), (f), and (h) in Figure 7 show observed (green line) and future (blue
and red lines) changes in the return period associated with maximum water level events due to sea-level rise. These curves were obtained by using, in Eq. 8, the MEVD with parameters estimated on 5-year sliding windows. As noted above, changes in return levels are nonlinear way: relative changes are more significant for smaller extremes than for larger ones. The $Tr$ vs. $z$ curves are concave downward and display varying slopes depending on the site explored. When a fixed return period is considered (e.g., 500 years), the mean sea level projections quantify the expected increase in extreme total water level peaks

**Table 4.**

Results of the percentage changes in total water level (Δz) obtained with the two future scenarios (RCP4.5 and RCP8.5) and the return periods (100 and 500 years) for the four sites under analysis.

| Tr (yrs) | RCP | Δz (%) | | | |
|---|---|---|---|---|---|
| | | Venice | Hornbæk | Marseille | Newlyn |
| 100 | 4.5 | 16.75% | 14.22% | 22.62% | 11.29% |
| | 8.5 | 22.82% | 21.76% | 29.73% | 15.26% |
| 500 | 4.5 | 14.60% | 11.70% | 16.24% | 11.23% |
| | 8.5 | 19.92% | 18.18% | 21.91% | 15.09% |

for that particular return period. These changes vary heterogeneously across the different coastal sites explored. By comparing the percentage changes associated with the two emission scenarios and the two return periods (Table 4), Venice and Marseille are seen to experience the greatest changes in extreme total water levels (e.g., with reference to $Tr = 100$ years and RCP8.5, the variations at Venice and Marseille sites are approximately 23% and 29% respectively). All sites display greater percent changes for the lower 100-year return period in each scenario, i.e. "less-infrequent" extremes will be most impacted by sea-level changes in the near future.

Changes in sea-level extremes can also be studied by focusing on changes in the return period of a fixed value of the total water level. To this end, one can define a sensitivity measure as:

$$SM = \frac{1}{Tr} \cdot \frac{dTr}{dmsl} = -\frac{1}{Tr} \cdot \frac{1}{[1 - G(z - msl)]^2} \cdot f(z - msl) = -f(z - msl) \cdot Tr \qquad (9)$$

which is obtained by derivation of Eq. 8, and where $f(z) = \frac{dG}{dz}$ is the probability density function associated with $G(z)$. Eq. 9 shows that, at a given site and for a set value of $z$, the relative change in return period grows linearly with $Tr$. For example, see in Figures 6b, d, f, h how, for a given value of $z$, changes (horizontal spacing between the curves) are greater for $Tr = 1000$ years than for $Tr = 500$ years. The expression for $SM$ also tells us that changes in $Tr$ are more significant, everything else being equal, for values of $z - msl$ near the mode of the distribution, where $f(z - msl)$ is maximum (e.g., compare changes at the Venice or Hornbæk sites with those at Newlyn for a same initial value of $Tr$). Finally, Eq. 9 shows that percentage changes in $Tr$ are highly site-dependent through the shape of $f(z - msl)$.

## 4 Conclusions

The comparative examination of extreme value distributions applied to observed sea levels at several sites along European coasts provides insights into the predictive performance of traditional and new approaches. Our analyses confirm some practical and conceptual advantages of the MEVD with respect to traditional methods. A cross-validation scheme (with 1,000 realizations for each sites) was used to compare model performance in high-quantile estimation. The use of two independent sub-sample (calibration and test sample) allows the quantification of actual predictive uncertainty.

We find that the MEVD approach provides reliable estimates of high quantiles for almost all the gauge stations explored,

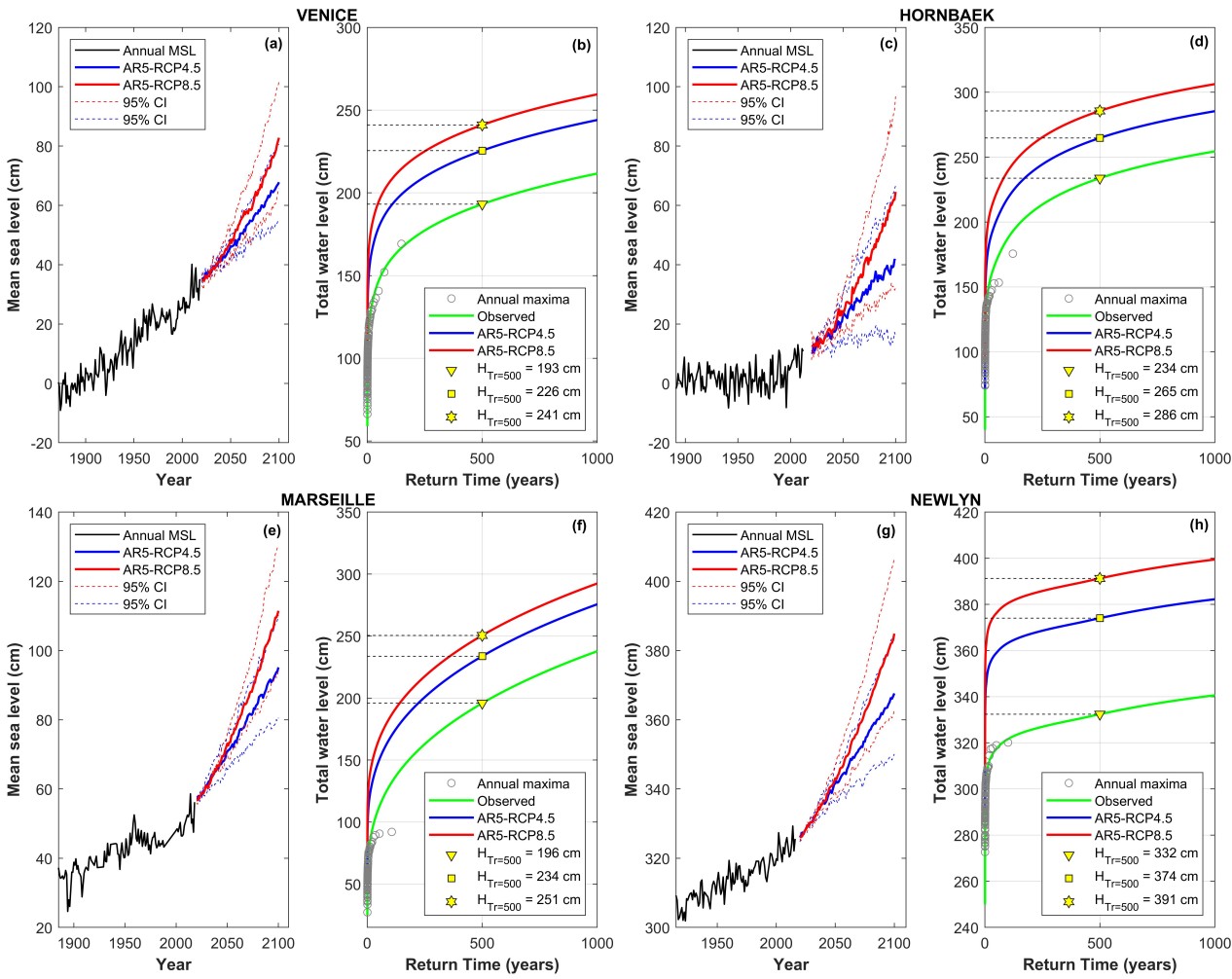

**Figure 7.** Future total water level projections, with respect to the current mean sea level, in: Venice (IT, panels (a) and (b)), Hornbæk (DK, panels (c) and (d)), Marseille (FR, panels (e) and (f)), and Newlyn (UK, panels (g) and (h)). The panels represent: 1) (a), (c), (e), and (g): annual (black line) and future mean sea level until 2100 with RCP4.5 (blue line) and RCP8.5 (red line). Dashed lines represent the 95% confidence intervals. 2) (b), (d), (f), and (h): return period curves for extreme total water level. The green curve represents the estimates obtained with the observed record; the blue and red curves represent the estimates obtained with the projected sea-level rise in the year 2100 with RCP4.5 (blue) and RCP8.5 (red) respectively; the grey dots indicate the observed annual maxima. Triangle, square and pentagon highlight the heights of extreme total water levels for a fixed return period equal to 500 years.

particularly when sufficiently long calibration sample sizes are considered. Differences in performance between the MEVD framework and GEV-based approaches are not large, and a definitive conclusion on an optimal solution independent of the return period of interest remains elusive. However, small differences in the estimation accuracy are relevant for engineering

applications when dealing with rare extreme events. If we focus on high-return period quantiles estimation, our analyses show that the MEVD approach provides reliable estimates for almost all the gauge stations explored. Data from the Marseille gauge station exhibit a behaviour that deviates from those from other sites, showing an inferior predictive performance of the MEVD with respect to GEV-based approaches. We interpret this fact to be linked to the small average number of sea-level peaks every year: the small sample of yearly ordinary events available prevents the MEVD from adding significant information with

respect to GEV-BM and POT-GPD. Conversely, when we evaluate methods performance for intermediate return period values, GEV-BM displays an overall greater robustness, and MEVD exhibit a greater absolute value of the estimation error. Unfortunately, the size of the available datasets does not allow to explore model performance for greater values of the return period. Future work could investigate if the estimation error can be reduced, with respect to what was found here, by using different approaches, e.g., by assuming "time-invariant" parameters in the ordinary distribution, whose estimation would thus

be performed on the entire calibration dataset, rather than on relatively short sliding windows. Synthetic water level time series may be produced by one of the several existing numerical models to extend analyses to arbitrarily long return period values. Finally, we explored projections of the frequency of extreme total water levels driven by changes in mean sea level. The sensitivity of extreme water level frequency to sea-level rise is location-dependent and we find that, at a given site and for a set value of the total water level extreme, the relative change in return time grows linearly with return time itself.

*Data availability.* All data used are publicly available from sources cited in the main text. Venice sea-level data were obtained from the "Centro Previsioni e Segnalazioni Maree" of the Venice Municipality (https://www.comune.venezia.it/it/content/centro-previsioni-e-segnalazioni-maree). The remaining water level data were downloaded from the University of Hawaii Sea Level Center (UHSLC) repository (http://uhslc.soest.hawaii.edu/data/?rq#uh745a/). The CMIP5 model outputs used for the future total water level projections are available at the "Integrated Climate Data Center" (ICDC) database of the University of Hamburg (https://icdc.cen.uni-hamburg.de/

en/ar5-slr.html).

*Author contributions.* M.M. designed and coordinated the research. M.F.C. performed the research and analyzed the data. Both Authors contributed to the writing and editing of the manuscript.

*Competing interests.* The authors declare that they have no conflict of interest.

*Acknowledgements.* M.F.C. acknowledges the PhD School in Civil Sciences and Environmental Engineering by University of Padova for funding her PhD. M.M. acknowledges financial support by the "Scientific activity performed with the contribution of the Provveditorato for the Public Works of Veneto, Trentino Alto Adige, and Friuli Venezia Giulia, provided through the concessionary of State Consorzio Venezia Nuova and coordinated by CORILA in the framework of the Venezia 2021 Research Program".

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
