# Peer review of "Extreme coastal water level estimation and projection: A comparison of statistical methods"

_Natural Hazards and Earth System Sciences, 2021_

## Author Comment (AC2)

**Authors' replies to Reviewer 2 comments for NHESS-2021-236**

We would like to thank Dr. Philip Ward and the two anonymous Reviewers for their helpful comments and suggestions, which much improved our manuscript. We appreciate the constructive comments received, which are discussed in detail. In general, some of the comments in particular, dealing with the way comparisons between extreme value distributions are done, prompted a shift in the focus of the manuscript. As Reviewer 2 puts it, the manuscript provides a comparison between possible approaches to extreme water level analysis. We have thus proposed a more balanced title, which does not needlessly over-emphasize the most recently proposed method and more objectively reflects the findings and the newly introduced analyses.

Below we provide our discussion of Reviewer's comments (in blue italic font) and describe the changes addressing them.

*Three approaches are compared in this research, i.e., GEV distribution on annual peak maxima (GEV-BM), the Metastatistical Extreme Value Distribution (MEVD), and GEV distribution on peaks over a higher threshold (GEV-POT). With respect to the latter approach, wouldn't it be better to rely on a Generalized Pareto Distribution (GPD) when threshold exceedances are considered? As far as I remember, GPD is a derivation of GEV for POT data; as such, is it conceptually correct to test a GEV distribution rather than a GPD on POT data? Please comment on this in the Methods section and/or extend the explanation in the Introduction (e.g. lines 37-39).*

We now realize that the original description of the traditional extreme value analysis methods was not entirely clear. We have renamed what was originally called GEV-POT as POT-GPD, to avoid the confusion pointed out by this Reviewer. In particular, we do use a GPD distribution for the selected events over a high threshold in the POT approach (please also see discussion on differences between POT-GPD and the MEVD approach used here). The Materials and Methods section in the revised manuscript now contains a more detailed description of both GEV-BM and POT-GPD methods that clarifies the above points.

*Lines 15-19 in the Introduction. As you speak of "active field" as for the modeling of extreme value probability of occurrence, you could reference more recent works.*

Thank you for the suggestion. We agree and the Introduction section in the revised manuscript now contains more recent works, e.g. the following:

a. Miniussi, A., and Marra, F.: Estimation of extreme daily precipitation return levels at-site and in ungauged locations using the simplified MEV approach. Journal of Hydrology, 603(B), 126946, https://doi.org/10.1016/j.jhydrol.2021.126946, 2021.

b. Mekonnen, K., Melesse, A. M., Woldesenbet, T. A.: Effect of temporal sampling mismatches between satellite rainfall estimates and rain gauge observations on modelling extreme rainfall in the Upper Awash Basin, Ethiopia. Journal of Hydrology, 598, 126467, https://doi.org/10.1016/j.jhydrol.2021.126467, 2021.

c. Cancelliere, A.: Non Stationary Analysis of Extreme Events. Water Resources Management, 31, 3097-3110, https://doi.org/10.1007/s11269-017-1724-4, 2017.

d. Elvidge, S., and Angling, M.J.: Using extreme value theory for determining the probability of Carrington-like solar flares. Space Weather, 16, 417-421. https://doi.org/10.1002/2017SW001727, 2018.

e. Rypkema, D.C., Horvitz, C.C., and Tuljapurkar, S.: How climate affects extreme events and hence ecological population models, Ecology, 100, 6, https://doi.org/10.1002/ecy.2684, 2019.

f. Chan, S., Chu, J., Zhang, Y., and Nadarajah, S.: An extreme value analysis of the tail relationships between returns and volumes for high frequency cryptocurrencies, Research in International Business and Finance, 59, 101541, https://doi.org/10.1016/j.ribaf.2021.101541, 2022.

*Line 29 in the Introduction. The list of reference is rather long; perhaps it would be enough to cite a few works and the "references therein".*
Thanks for the suggestion.

*Line 48 in the Introduction. You can also cite Solari et al. (2017).*
Thanks, Solari et al. (2017) is now cited in the revised manuscript.

*Page 4, Fig. 1. Please reduce the y-axis range for Marseille plot.*
Agreed. The y-axis range for Marseille was reduced as shown in the following revised Figure 1.

[Figure]

*Figure 1. Revised Figure 1.*

*Page 5, line 107. If I understood correctly, "year" in the following line should be replaced with "block".*
Thanks for this careful correction. It is correct, in the revised manuscript we have changed "year" with "block".

*I would swap Section 2.2.1 and Section 2.2.2. First explain how you pre-processed the data, then the distribution used to model them.*
Agreed.

*Section 2.2.1. I think you should explain what are the cumulative distributions F you tested for the ordinary values, and which one did you choose.*
In the revised manuscript we have clarified the potential distributions that were tested for sea level frequency analysis (i.e. Gamma, Weibull and Generalized Pareto distributions). Based on the comparative evaluation of the performance of these three probability distributions, described in the

revised supplementary materials, the Generalized Pareto Distribution emerged as the best model for the "ordinary" water level peaks (see revised Section 3.2, lines 264-268).

*Section 2.2.2, line 134. The fact that you neglect the interactions between tides and surges means that gauges are placed in deep waters. Is that true? Please add the respective water depths in Table 1 if such info are available.*

This statement may have generated some confusion and needs additional discussion. We now clarify that the tide-surge interaction is significant and needs to be taken into account when the surge and tide components are studied separately.

The revised manuscript at line 134 now explains that: "However, this effect is significant and needs to be taken into account when the surge and tide components are studied separately. Since here we do not attempt to separate these contributions but only analyze the sum given by the combination of the water level setup, induced by meteorological forcing, and the astronomical tide, hereafter we will neglect their non-linear interactions and will consider the observed sea level as the sum of additive components".

*Section 2.2.2. Please number the equations.*

Thank you, agreed.

*Section 2.2.3, lines 171-173. This paragraph is unclear. Indeed, looking at the correlograms (Fig. S1) it seems that independent events are achieved for no lags. This aspect is crucial so it should be better explained. Correlograms also reveal that tides are relevant (negligible) in Venice and Newly (Marseille and Hornbaek). Perhaps you could comment on this in the paper.*

Thank you for this comment. In the Supporting Information, we have added Figure S2. The revised manuscript now discusses (lines 171-173) that: "The analysis of the correlograms of selected water level peaks shows that some correlation persists also for long time lags and also in the de-clustered time series. Even though the strength of this correlation is relatively small (the ACF is always less than 0.3) this periodic correlation should be considered in the interpretation of subsequent results, as it may impact the performance of statistical modelling. The de-clustering process does significantly decrease correlation as may be seen by comparing Figure S1 (ACF prior to de-clustering) and Figure S2(after de-clustering). Interestingly, it is seen that the tidal contribution (that generates periodicities in the ACF) is strongly visible in Venice and Newlyn, while it is quite small in Hornbaek and Marseille. However, after de-clustering, the residual correlation is associated with the tidal component also in Hornbæk and Marseille. The existing literature, which focuses on the storm-surge component only, uses shorter time lag values due to the shorter correlation of the surge component due to atmospheric drivers. For example, the independence…".

[Figure]

*Figure S2. Correlograms for independent daily maxima water levels obtained by dividing the time series (of independent events) into 30-day bins.*

*Section 2.2.3, line 177. Please use consistent tenses throughout the paper when referencing other works. For instance, here you say "Bernardara et al. (2011) adopted", while previously you use the present tense (e.g. page 6, line 117 or later in the paper at page 8, line 212).*
Thank you, we have revised the tenses throughout the manuscript.

*Page 9, line 239. I do not understand why the return period is expressed for annual maxima (AM). Apologies but I am not familiar with the MEVD, however it is clear that it allows to select multiple events per year. Then, why Eq. (3) is defined with respect to AM data?*
Exactly, the MEVD uses more observations to estimate the parameters of the distribution of "ordinary values". However, the MEVD cumulative distribution (Eq. (2)) is still the distribution of the annual maxima (line 100-103), estimated using a much greater sample than just yearly maxima.

*It is not clear the purpose of Section 3.1, given that no non-stationary distributions are subsequently employed. However, if you want to keep it, I suggest adding the confidence intervals of the slopes fitted to the data (and perhaps comment them with respect to the p-values of the Mann-Kendall test).*
Section 3.1 is a preliminary analysis of the extreme sea levels to understand whether long-term trends, unrelated to sea-level rise but to other factor (e.g., human-induced factors, morphological variations, etc.), are highlighted in the "cleaned up" signal (i.e. without mean sea level) of long time series of sea-level observation considered in this application. As suggested by this Reviewer, we have added confidence intervals in Figure 2.

*Section 3.2, lines 277-279. What does it mean that thresholds are selected "based on local tidal ranges"? This is a pivotal step of the study, please extend the explanation (you could also add it to the Methods section).*
Tidal range is defined as the sea level elevation between high and low water level over a tidal cycle. Differences in tidal range are often important, and are related to variations in coastal processes and

morphology. Tidal range varies between locations and time scales according to the hydrodynamic response of a particular bay or estuary to astronomical tidal forcing. Since the choice of the threshold is site-specific, we must consider this parameter in its selection.

As highlighted from the Reviewer 1, we have moved the lines 276-277 to line 121. The revised manuscript now discusses here that: "The threshold is set to be large enough to filter out water level peaks that are likely to be associated to conditions without any storm contribution and sufficiently low to maximize the amount of information used. In addition to the above, we choose the threshold value that produces the minimum estimation error under the MEVD framework".

*Figures S2-S6. Please use a 1:1 axis ratio. This would help to assess the quality of the fit.*
Thank you, agreed.

*Figure 5. Levels in the return period plots refer to z or h? I find the terminology rather confusing throughout the whole manuscript, e.g. sometimes you talk about storm surge, some other time about extreme sea levels. Please be consistent.*
We apologize for the lack of clarity. It is *z*. To avoid confusion, in Figure 5 we have replaced "water level" with "total water level" (i.e. the variable *z*). In the revised manuscript, we have revised all the notation and terms used.

*Conclusions, line 349. Please specify that MEVD outperforms the other distributions for long enough calibration periods.*
Thank you for the suggestion. We have revised the sentence accordingly.

---

## Author Response (AR1)

**Authors' replies to Reviewers comments for NHESS-2021-236**

We would like to thank Dr. Philip Ward and the two anonymous Reviewers for their helpful comments and suggestions, which much improved the manuscript. We appreciate the constructive comments received, which are discussed in detail. In general, some of the comments dealing with the way comparisons between extreme value distributions are performed, prompted a shift in the focus of the manuscript. As Reviewer 2 puts it, the manuscript provides a comparison between possible approaches to extreme water level analysis. We have thus proposed a more balanced title, which does not needlessly over-emphasize the most recently proposed method and more objectively reflects the findings and the newly introduced analyses.

Below we provide our discussion of Reviewer's comments (in blue italic font) and describe the changes addressing them (in black font).

**Reviewer #1:**

*From the title of the manuscript, the reader expects to read a study about extreme storm surge. However, the study's objectives (Lines 66-68) refer to extreme sea level. Later on, Line 150, the Authors say that they will investigate the variable h(t) being the sum of tide and storm surge, so sea-level without mean sea level. I would encourage the Authors to clearly state the variable of interest and the variable used when performing the analyses, see also other comments below.*

Thank you for this suggestion. To avoid any confusion about the focus of our work, we have changed the title of the manuscript to "*Extreme coastal water level estimation and projection: A comparison of statistical methods*". Throughout the manuscript we now use the term "coastal water level" when referring to the sum of the tide and surge components.

*Information regarding MEVD, which is the main method investigated in the manuscript, is limited. The Authors say that this method guarantees "the least amount of a-priori assumption" (line 56). However, the following assumption must be made: F(x,θ) in Eq. 2, the threshold for the ordinary values, the estimation window for parameter estimation, the time-lag to ensure independence between ordinary values. How then is this method the one with the least amount of a-priori assumptions? I suggest clarifying further the advantages of the MEVD compared to the other two methods investigated.*

In the revised manuscript, we have expanded the description of the MEVD and we better emphasize the differences between the MEVD and traditional approaches. In particular, we now clarify, at line70, that: "Moreover, the MEVD framework (i) is a non-asymptotic extreme value distribution, which does not require the number of events/year to be large as in the traditional theory, and (ii) makes no a-priori assumptions on the properties of the event occurrence process (while, e.g. POT-GPD assumes a Poisson occurrence process)".

*Moreover, additional information should be discussed: how the threshold for the ordinary value was selected (line 121 says "as small as possible").*

We have moved the discussion about threshold selection, previously at lines 276-277, to line 216, to clarify. The revised manuscript now discusses, at lines 213-218, that: "This threshold is chosen to be as small as possible (differently from the POT approach), to retain as much of the observational information as possible, and will be dependent on the magnitude of the local tidal range (sea level difference between high and low water level over a tidal cycle), and of storm contributions. Additionally, the threshold is set to be large enough to filter out water level peaks that are likely fully determined by tidal fluctuations, in the absence of any storm contribution. Given the above constraints, we also choose the threshold value that minimizes the estimation error under the MEVD framework".

*How the 5-year estimation window was selected.*

The revised manuscript now includes, at line 220 a discussion of the optimal estimation window length: "In the present application, the optimal estimation window length was set to 5 years to obtain a more robust parameters estimation, especially when few values in each year are available".

*Why the 30-day lag time for the independence of the ordinary value is so different compared to the values found in the literature (lines 173-179).*
We agree that the wording in the previous manuscript generated some confusion on this point. The previous literature introduces minimum time lags with different objectives. Here a minimum time lag separation is introduced to eliminate correlations between water level peaks induced by the deterministic tidal contribution, which has a long periodicity linked to the main lunar cycle of about 28 days. The existing literature, on the other hand, focuses on the storm-surge component only and thus uses shorter time lag values due to the shorter correlation of the surge component due to atmospheric drivers. The revised manuscript now clarifies, at line 278, that: "We note that the literature implementing de-clustering approaches to coastal level signals normally focuses on studying the storm-surge component only. As result it uses threshold time lag values that are smaller than those adopted here because characteristic correlation times of the surge component are significantly smaller than those associated with the sum given by the combination of surge and tidal components. For example, the independence […]".

*How F(x,θ), which turns out to be a GDP (Line 267), is different compared to the classical GDP.*
We have renamed what was originally called GEV-POT as POT-GPD, to avoid the confusion pointed out by this Reviewer.
We interpret this question as asking how the MEVD is different from the POT-GPD methods, which also uses a GPD distribution. The difference is that the GPD used in the MEVD approach aims to capture the distribution of all the ordinary values (i.e. those in the main body of the probability distribution), obtained by imposing a "low threshold". On the contrary, the classic POT-GPD method adopts a very high threshold to select independent large events in the tail of the distribution. Additionally, the MEVD does not require one to assume that event arrival is Poisson-distributed. Zorzetto et al. (2016) highlighted that if one assumes (i) $x$ to be the excess over a high threshold, (ii) $F(x; \theta)$ to be a Generalized Pareto Distribution (with fixed, deterministic parameters), and (iii) $n$ to be generated by a Poisson distribution, then the GEV distribution is recovered as a particular case of the MEVD by means of the POT approach. We now briefly clarify this point in the revised manuscript (lines 408-411) as follows: "We highlight that the GPD used in the MEVD framework is obtained by imposing a small threshold (differently from the high threshold adopted in the POT-GPD approach) to capture the distribution of the main body of the probability distribution of the ordinary events and does require the event arrival process to be Poisson (Marani and Zorzetto, 2019)".

*I do see the value in implementing the cross-validation procedure to assess the predictability power of the distribution selected as representative of the observations. At the same time, I see the cross-validation as an additional measure of goodness of fit rather than the main one.*
We respectfully disagree on the statement regarding out-of-sample vs in-sample tests. Cross-validation is not a measure of goodness of fit. Comparisons of estimation outcomes against independent data (not used in calibration) quantify how a statistical model can predict the likelihood of the "next event" and gauge how inferences from a statistical method can capture the properties of the underlying physical process, as opposed to just describe the specific dataset on which it is calibrated.

*The NDE only tests if the one quantile associated with the return period Tr of interest is well captured. What about the other quantiles? Is the distribution representative of the entire sample?*
We are grateful to this reviewer for this useful comment. The NDE represents an average measure (average among the realizations $p = 1, ..., Nr$ where $Nr = 1000$ in this application) of a standardized distance between estimated and empirical quantiles, and can be computed for any

return period. We had focused on the highest quantile, estimated from the independent test dataset, because its estimation is the most uncertain and the most valuable in practice. However, we agree with the Reviewer that it is useful to ask the question "is the extreme value distribution representative of the entire range of return times of interest?". To this end, we have performed additional analyses to evaluate methods performance also for intermediate $Tr$ values, greater than the calibration sample size (for $Tr<S$ empirical quantiles can be used, with little need of distribution fitting). The results are reported in the Figure below (obtained by estimating the probability distribution parameters on 30-year calibration sub-samples).

[Figure]

*Figure R1 (Figure 6 in the revised manuscript): Median of the non-dimensional error (NDE) for return period greater than the calibration sample size in test sub-sample for the GEV-BM, POT-GPD, and MEVD approaches (magenta, blue and green dots respectively). The results are obtained for the Venice (IT), Hornbæk (DK), and Newlyn (UK) sites and by estimating the distribution parameters on 30-year calibration sub-samples.*

The new analyses suggest that when we focus on the median error associated with moderate values of the return period, GEV-BM displays an overall greater robustness (e.g., in the case of Venice and Hornbæk sites) with respect to POT-GPD and MEVD, which exhibit greater fluctuations. In particular, results show that MEVD is a good model for the highest values of the return period, but exhibit a greater absolute value of the estimation error for smaller $Tr$. The size of the available datasets does not allow to explore what happens for even greater values of $Tr$. Future work could investigate if estimation error can be reduced using different approaches to parameter estimation (e.g., by assuming "time-invariant" parameters in the ordinary distribution, whose estimation would thus be performed on the entire calibration dataset, rather than on relatively short sliding windows) and compute NDE for much greater values of $Tr$ based on synthetic water level time series (which may be as long as is desired) produced by one of the several existing numerical models. We have now added this Figure to the manuscript, along with the above discussion. This addition, stemming from this Reviewer's suggestion, also prompted a slight change of title, to reflect that the focus is a comparison of methods and to avoid suggesting that the MEVD is necessarily superior at all $Trs$.

The revised manuscript discusses at lines 472-487 that:

"We also provide a comparative analysis between the three methods to evaluate if the tested extreme value distributions are representative of the entire range of return times of interest. To achieve this purpose, we evaluate methods performance also for intermediate $Tr$ values, greater than the calibration sample size, since for $Tr < S$ the empirical quantiles can be used. We perform this additional analysis for the Venice, Hornbæk and Newlyn sites. Figure 6 summarizes the results obtained by estimating the probability distribution parameters on 30-year calibration sub-samples. The analyses suggest that when we focus on the median error associated with moderate values of the return period, GEV-BM displays an overall greater robustness (e.g., in the case of Venice and Hornbæk sites) with respect to POT-GPD and MEVD, which exhibit greater fluctuations. In particular, results show that MEVD is a good model for the highest values of the return period, but exhibit a greater absolute value of the estimation error for smaller $Tr$. Overall, the results suggest that no single approach is clearly superior at all values of $Tr$, due to a large variability in the estimates. For example, for the Venice site there is a decrease (in many cases an unbiased estimates) in the MEVD NDE values for intermediate $Tr$ (between 85 and 105 years) while for greater $Tr$ values (but smaller than $Tr_{max}$) the error shows an overestimation of the actual quantile with respect to traditional approaches (which exhibit an underestimation tendency). To be more specific, if $Tr > 105$ years are considered, MEVD yields error estimates between zero and <10%, while errors associated with GEV-BM and POT-GPD lie between zero and <-20%. The Hornbæk site shows similar results to the Venice site, while Newlyn's results exhibit more fluctuations for large $Tr$ values with much reduced smaller amplitudes and values of the NDE".

*Also, how the observed quantile h(obs,p) is calculated? Which sample (M, S, or V) is used? The Q-Q plots are mentioned only in the results section and they are only performed for the 30 years in-sample test. In my opinion, the Q-Q plots put the NDE into perspective and should be included as goodness-of-fit method. Also, it would be useful to have them in the main manuscript. I do understand that the space is limited, maybe the Authors could consider including in the main manuscript only the ones related to the MEDV.*

We have added the following description (lines 309-312): "As usual in frequency analysis, we associate to each observed yearly maximum, $x_i$, an empirical frequency value given by Weibull's estimator $F_i = \frac{i}{(V+1)}$, where $i$ is the rank of $x_i$ in the list of yearly maxima sorted in ascending order, and $V = M - S$ is the sample size in the validation sub-sample. The return period $Tr$ associated with each yearly maximum is then simply $Tr_i = \frac{1}{1-F_i}$."

We have now clarified that the test sub-sample ($V$) is used to extract the empirical quantiles to compare them with estimated ones (line 313). Regarding the Q-Q plots, we agree with the Reviewer and we have included in the main text the QQ-plots related to the MEVD (see new Figure 3 in the revised manuscript).

*In the section Return Period, the definition of Equation 4 needs to be further discussed. Even if the Authors replace (h) with (z-msl), Equation 4 is still the return period of (h), and not the return period of the (z), as indicated by the Authors. Mean sea level (msl) shows a clear linear trend and such trend is recognizable in (z). Similarly, in Equation 5, the distribution G is the distribution of the variable (h) and not the variable (z) as reported in line 341. This has an implication in Figure 5. I assume that the y-axis in Figure 5 "water level" refers to the variable (z). This variable (z) is time-dependent, while in Figure 5 it seems like the statistical properties of (z) are constant. I would have expected something similar to the effective return level plots, to show the effect of sea-level rise. How (msl), which is time-dependent, is added to (h), which is not time-dependent, to derive Figure 5? I suggest clarifying the transition from the analysis on the variable (h), a random variable, to (z), which presents a linear trend due to (msl). I also suggest being more precise with the notation and the terms used throughout the manuscript. It is very difficult to understand the variables the Authors refer to because are often called with many different terms, e.g., total sea level, water level, extreme sea level...*

We apologize for the lack of clarity. In the revised manuscript, we have revised all the notation and terms used. In particular, we have indicated the variable *z* as "total water level" and *h* as "coastal water level". To avoid confusion, in Figure 7 (previously Figure 5) we have replaced "water level" with "total water level" (i.e. the variable *z*).

Regarding the comments on $G(h)$, we see that some confusion may have arisen. We have now clarified how the exceedance distribution of variable $H$ (coastal water level) is the same as the exceedance distribution of $Z$ (total water level) by expanding the existing discussion (formerly lines 240-242), which now reads (lines 365-369): "Because for a fixed value of mean sea level there is a one-to-one relation between the value of the sum of the astronomical and the storm surge contribution, $h$, and the total water level, $z = h + msl$, one can write $G_h(h) = P[H > h] = P[H > z - msl] = P[Z - msl > z - msl] = P[Z > z] = G_z(h)$, such that Eq. 7 can be used, once the cumulative distribution is known and for each (time-dependent) value of *msl*, to determine the return period of the total water level (at the time when *msl* is evaluated): $Tr(z) = 1/1 - G_z(h)$".

This equality is independent of the fact that *msl* may change over time as the one-to-one relation between $H$ and $Z$ holds at all times. The possibility of projecting probability distributions and return periods into the future precisely depends on the fact that we need only substitute updated values of $msl(t)$ to infer the probability of future extreme total water levels.

*The Authors say that "MEVD proves to be a good model for the extreme sea levels" (line 288) and that "MEVD-based estimates outperform the traditional approaches" (line 301). I do fail to see what the Authors describe. In the QQ-plots Figure S2-6, MEVD in the in-sample analysis has, in general, the highest variability, especially compared to the GEV. In the out-of-sample, MEVD looks better for lower quantiles, but it has quite a large variability for higher quantiles, compared to the other distributions. Overall, it is difficult to quantify which distribution performs best. This is also reflected in the NDE plots, Figure 3, where the differences between distributions are minimal.*

We agree with this Reviewer that differences in performance are not large between MEVD and GEV, and the revised manuscript now does not draw a definitive conclusion on which approach is best independently from the return period of focus. However, small differences in the estimation accuracy are relevant for engineering applications when dealing with rare extreme events. Figure 4 (previously Figure 3) shows a better performance by MEVD for the largest quantile for all sites except Marseille. We thus argue that the improved performance of MEVD for large *Tr* may have a significant impact on the effective design of coastal defense structures (e.g., see Table 3 and Figure 4(a), (b) and (c)). The additional graph produced above to answer previous comments by this Reviewer shows small differences in the estimation accuracy of different approaches at different sites. The revised manuscript now reports at lines 480-487: "Overall, the results suggest that no single approach is clearly superior at all values of *Tr*, due to a large variability in the estimates. For example, for the Venice site there is a decrease (in many cases an unbiased estimates) in MEVD NDE values for intermediate *Tr* (between 85 and 105 years) while for greater *Tr* values (but smaller than $Tr_{max}$) the error shows an overestimation of the actual quantile with respect to traditional approaches (which exhibit an underestimation tendency). To be more specific, if *Tr* > 105 years are considered, MEVD yields error estimates between zero and <10%, while errors associated with GEV-BM and POT-GPD lie between zero and <-20%. The Hornbæk site shows similar results to the Venice site, while Newlyn's results exhibit more fluctuations for large *Tr* values with much reduced smaller amplitudes and values of the NDE".

**Point by point comments:**

*Line 92. Please revise the notation. Pr(Mn<= x) = F(x)^n where Mn is the maximum of a sequence of independent random variable X. See also Coles 2001 (line 415)*

Agreed, we have changed the notation accordingly.

*Line 154. Additional discussion is needed concerning the fact that h(t) can be considered a stochastic variable even though a determinist component is included. Also, a literature review on indirect and direct methods (Line 149) for extreme sea level is missing.*

Thanks for the suggestions. We have improved the discussion concerning the two aspects highlighted from the Reviewer. In particular, the revised manuscript, at lines 123-130, now reads: "Two classes of methods are widely used to estimate the probability of occurrence of extreme sea levels: direct and indirect methods. Indirect methods model separately the deterministic and the stochastic components of z(t) then recombined by convolution. Examples are the joint probability method (Pugh and Vassie, 1979, 1980), the revised joint probability method (Tawn and Vassie, 1989), the exceedance probability method (Middleton and Thompson, 1986; Hamon and Middleton, 1989), and the empirical simulation technique (Scheffner et al., 1996; Goring et al., 2011). Direct methods, such as the one adopted here, analyze observed values compounding the astronomical and stochastic storm-surge component. Direct methods mostly differ based on the analysis approach adopted, such as the annual maxima (Jenkinson, 1955; Gumbel, 1958), the peaks-over-threshold method (Davison and Smith, 1990), or the r-largest method (Smith, 1986; Tawn, 1988)]. Here, we study [...]".

*Lines 133. The Authors discuss the negligibility of tide-surge interaction. Does this condition hold in the case of Punta della Salute which is located within the Venice Lagoon?*

The statement mentioned by the Reviewer may have generated some confusion and needed additional discussion. We now clarify that the tide-surge interaction needs to be understood and quantified when attempting to separate the surge and tide components.

The revised manuscript now explains, at lines 106-109, that: "However, this effect needs to be taken into account when separating the surge and tide components. Here, we do not attempt to separate these contributions, but we only analyze the sum given by the combination of the water level setup, induced by meteorological forcing, and the astronomical tide. Hence we simply study such sum as the final result of the non-linear interactions between individual components".

*How the GDP threshold is selected and tested?*

As described from lines 272 to 275 (lines 415-417 in the revised manuscript), the optimal GPD threshold value was determined by studying the stability of the GPD shape and modified scale parameters. To evaluate the goodness of fit of the distribution with respect to different threshold values, diagnostic graphical plots were constructed.

In the discussion to a previous comment by Reviewer#2 we clarify that we have moved the discussion about threshold selection (in the case of ordinary sample), previously at lines 276-277, to line 216. The revised manuscript now discusses, at lines 213-218, that: "This threshold is chosen to be as small as possible (differently from the POT approach), to retain as much of the observational information as possible, and will be dependent on the magnitude of the local tidal range (sea level difference between high and low water level over a tidal cycle), and of storm contributions. Additionally, the threshold is set to be large enough to filter out water level peaks that are likely fully determined by tidal fluctuations, in the absence of any storm contribution. Given the above constraints, we also choose the threshold value that minimizes the estimation error under the MEVD framework".

*It would be very interesting and useful to appreciate the difference between the performance of the distribution functions to see the sample of maxima used for fitting the distributions.*

We think the Reviewer is highlighting the need to include, in addition to Table 2, a comparative figure between the extreme time series used for fitting the distributions. The Supporting Information now includes this additional figure (Figure S3) that displays the sample of maxima used to infer the distributions, i.e. annual maxima (GEV-BM), exceedances over the threshold (POT-GPD) and ordinary values (MEVD).

*Lines 205-209. My suggestion is to revise this paragraph. The terminology is confusing. I believe the Authors here are discussing the variable (z), in which storm surge is a component.*
Thank you, agreed. The revised manuscript (lines 319-324) now reports: "Future increases in the frequency of extreme sea levels due to climate change will have serious impacts on coastal regions. These impacts will vary temporally and regionally, depending on (i) the local relative mean sea-level rise (including possible subsidence or uplift), (ii) current storm-surge intensity probability distributions, and (iii) changes in the dominant meteorological dynamics. In this particular application to extreme coastal water levels (i.e. the sum given by the combination of the water level setup, induced by meteorological forcing, and the astronomical tide), only the first two factors are considered".

*Lines 220-221. The Authors say that the tidal and storm components do not change over time as mean sea level. How did the Author check that no trend is detected in the variable h?*
The initial part of the results discussion illustrates a trend analysis of maximum yearly departures from the concurrent mean sea level (see Figure 2), whose results are consistent with the absence of trends at three of the sites considered, whereas trends at the remaining site (Venice) seem to be significant, though small. This finding supports the assumption that the sum of tidal and storm component does not change over time, which also finds confirmation in previous studies of past and future changes in extreme high water levels (e.g., Zhang et al., 2000; Woodworth and Blackman, 2004; Menéndez and Woodworth, 2010; Lowe et al, 2010; Haigh et al., 2014; Wahl et al., 2017). According to this literature, it is reasonable to assume that increases in extreme high sea levels are primarily a due to a rise in mean sea level. This implies that variations in storm activity (e.g. magnitude, trajectories and frequency) are comparatively smaller than future rise in mean sea level at most locations. Finally, the assumption is also confirmed in the IPCC AR5 report, which states that there is "low confidence" in region-specific projections of storminess and associated storm surges.

To clarify these points, the revised manuscript (lines 329-340) now reports: "Various techniques have been used to study possible changes in coastal flooding hazard (e.g., McInnes et al., 2013; Vousdoukas et al., 2016). Several authors have found that past variations in the frequency of occurrence of extreme sea levels have been primarily determined by changes in mean sea level (e.g., Zhang et al., 2000; Woodworth and Blackman, 2004; Lowe et al., 2010; Menéndez and Woodworth, 2010; Church et al., 2013; Haigh et al., 2014b; Wahl et al., 2017). This implies that effects of variations in storminess (e.g., magnitude, trajectories and frequency) have been small in the observational record compared to the dominant effects of mean sea-level changes (Haigh et al., 2014a). This notion is also confirmed by our trend analyses (see §3.1), which fail to detect trends in the maximum difference between total sea level and concurrent mean sea level except at one of the sites (Venice), where it is smaller (0.7 mm/yr) than past and projected rates of sea-level rise (respectively ~3.0 mm/yr and ~8.0 mm/yr at the end of the century, according to the RCP8.5 IPCC scenario). Based on these elements, here we estimate the probability of future total water levels along European coastlines by assuming that changes in the tidal and storm-surge components are negligible with respect to changes in mean sea-level, an assumption common to previous approaches (Araújo and Pugh, 2008; Haigh et al., 2010; Tebaldi et al., 2012)."

*Section 3: Was the trend test performed only on the annual maxima or also on the samples of maxima used to compute the GPD and the MEVD?*
The trend test was performed only on the annual maxima. The revised manuscript now clarifies at line 380 that: "To answer this question, in this work we focus on the deviation of the yearly maxima from yearly mean sea level and test for the presence of trend by the two-tail Mann-Kendall test".

*Line 281: Storm surge or storm surge and tide?*

Thank you for pointing this lack of clarity. It is storm surge and tide. In the revised manuscript, we have revised all the notation and terms used. The variable defined as the sum between surge and tide is now indicated as "*coastal water level*".

*Line 285: what is L?*

Thank you for catching this. We apologize for the mistake. "L" has been replaced with "M" which is the correct symbol used to indicate the time series length (as indicated in line 308, previously line 199).

**Reviewer #2:**

*Three approaches are compared in this research, i.e., GEV distribution on annual peak maxima (GEV-BM), the Metastatistical Extreme Value Distribution (MEVD), and GEV distribution on peaks over a higher threshold (GEV-POT). With respect to the latter approach, wouldn't it be better to rely on a Generalized Pareto Distribution (GPD) when threshold exceedances are considered? As far as I remember, GPD is a derivation of GEV for POT data; as such, is it conceptually correct to test a GEV distribution rather than a GPD on POT data? Please comment on this in the Methods section and/or extend the explanation in the Introduction (e.g. lines 37-39).*

We now realize that the original description of the traditional extreme value analysis methods was not sufficiently clear. We have renamed what was originally called GEV-POT as POT-GPD, to avoid the confusion pointed out by this Reviewer. In particular, we do use a GPD distribution for the selected events over a high threshold in the POT approach (please also see discussion on differences between POT-GPD and the MEVD approach used here). The Materials and Methods section in the revised manuscript (lines 144-172) now contains a more detailed description of both GEV-BM and POT-GPD methods that clarifies the above points.

*Lines 15-19 in the Introduction. As you speak of "active field" as for the modeling of extreme value probability of occurrence, you could reference more recent works.*

We agree, and the Introduction section in the revised manuscript is now updated to contain more recent works, e.g. the following:

a. Miniussi, A., and Marra, F.: Estimation of extreme daily precipitation return levels at-site and in ungauged locations using the simplified MEV approach. Journal of Hydrology, 603(B), 126946, https://doi.org/10.1016/j.jhydrol.2021.126946, 2021.

b. Mekonnen, K., Melesse, A. M., Woldesenbet, T. A.: Effect of temporal sampling mismatches between satellite rainfall estimates and rain gauge observations on modelling extreme rainfall in the Upper Awash Basin, Ethiopia. Journal of Hydrology, 598, 126467, https://doi.org/10.1016/j.jhydrol.2021.126467, 2021.

c. Cancelliere, A.: Non Stationary Analysis of Extreme Events. Water Resources Management, 31, 3097-3110, https://doi.org/10.1007/s11269-017-1724-4, 2017.

d. Elvidge, S., and Angling, M.J.: Using extreme value theory for determining the probability of Carrington-like solar flares. Space Weather, 16, 417-421. https://doi.org/10.1002/2017SW001727, 2018.

e. Rypkema, D.C., Horvitz, C.C., and Tuljapurkar, S.: How climate affects extreme events and hence ecological population models, Ecology, 100, 6, https://doi.org/10.1002/ecy.2684, 2019.

f. Chan, S., Chu, J., Zhang, Y., and Nadarajah, S.: An extreme value analysis of the tail relationships between returns and volumes for high frequency cryptocurrencies, Research in International Business and Finance, 59, 101541, https://doi.org/10.1016/j.ribaf.2021.101541, 2022.

*Line 29 in the Introduction. The list of reference is rather long; perhaps it would be enough to cite a few works and the "references therein".*

Line 35 now reports the following references: "(Fréchet, 1927; Dalrymple, 1960; Coles, 2001; Woodworth and Blackman, 2002; Hamdi et al., 2014, 2015, and references therein)".

*Line 48 in the Introduction. You can also cite Solari et al. (2017).*

Thanks, Solari et al. (2017) is now cited in the revised manuscript.

*Page 4, Fig. 1. Please reduce the y-axis range for Marseille plot.*
*A*greed. The y-axis range for Marseille was reduced as shown in the following revised Figure 1.

[Figure]

*Figure R2 (Figure 1 in the revised manuscript): Daily maximum sea levels at different gauge stations explored after pre-processing: Venice (IT), Hornbæk (DK), Marseille (FR), and Newlyn (UK).*

*Page 5, line 107. If I understood correctly, "year" in the following line should be replaced with "block".*
Thanks for this careful correction. It is correct, in the revised manuscript we have changed "year" with "block".

*I would swap Section 2.2.1 and Section 2.2.2. First explain how you pre-processed the data, then the distribution used to model them.*
Agreed.

*Section 2.2.1. I think you should explain what are the cumulative distributions F you tested for the ordinary values, and which one did you choose.*
In the revised manuscript we have clarified the potential distributions that were tested for sea level frequency analysis (i.e. the Gamma, Weibull and Generalized Pareto distributions). Based on the comparative evaluation of the performance of these three probability distributions, described in the revised supplementary materials, the Generalized Pareto Distribution emerged as the best model for the "ordinary" water level peaks. The revised Section 2.2.3, at lines 208-211, now reads: "In this particular application to extreme coastal water levels, three candidate probability distributions for $F(x;\theta_j)$ in Eq. 6 are tested, i.e. the Gamma, Weibull and Generalized Pareto distributions. Based on the comparative evaluation of the performance of these distributions, the Generalized Pareto distribution emerged as the best model for the "ordinary" coastal water level values.".

*Section 2.2.2, line 134. The fact that you neglect the interactions between tides and surges means that gauges are placed in deep waters. Is that true? Please add the respective water depths in Table 1 if such info are available.*

This statement has generated some confusion and needed additional discussion. We now clarify that the tide-surge interaction needs to be understood and quantified when attempting to separate the surge and tide components.

The revised manuscript now explains, at lines 106-109, that: "However, this effect needs to be taken into account when separating the surge and tide components. Here, we do not attempt to separate these contributions, but we only analyze the sum given by the combination of the water level setup, induced by meteorological forcing, and the astronomical tide. Hence we simply study such sum as the final result of the non-linear interactions between individual components".

*Section 2.2.2. Please number the equations.*
Yes, we agree with the suggestion and have numbered all equations.

*Section 2.2.3, lines 171-173. This paragraph is unclear. Indeed, looking at the correlograms (Fig. S1) it seems that independent events are achieved for no lags. This aspect is crucial so it should be better explained. Correlograms also reveal that tides are relevant (negligible) in Venice and Newly (Marseille and Hornbaek). Perhaps you could comment on this in the paper.*
We have added Figure S2 to the Supporting Information. The revised manuscript now discusses (lines 271-282) that: "The analysis of the correlograms of selected water level peaks shows that some correlation persists also for long time lags and also in the de-clustered time series. Even though the strength of this correlation is relatively small (the ACF is always less than 0.3), it could impact the ability of the MEVD, which assumes independence, to capture observed extreme behaviour. The de-clustering process does significantly decrease correlation, as may be seen by comparing Figure S1 (ACF prior to de-clustering) and Figure S2(after de-clustering). Interestingly, it is seen that the tidal contribution (that generates periodicities in the ACF) is strongly visible in Venice and Newlyn, while it is quite small in Hornbæk and Marseille. The underlying tidally-induced correlation becomes more clearly visible after de-clustering also in Hornbæk and Marseille. We note that the literature implementing de-clustering approaches to coastal level signals normally focuses on studying the storm-surge component only. As result it uses threshold time lag values that are smaller than those adopted here because characteristic correlation times of the surge component are significantly smaller than those associated with the sum between surge and tidal components. For example, the independence […]".

[Figure]

*Figure R3 (Figure S2 in the revised supplementary material): Correlogram plots for independent daily maxima coastal water levels with threshold lag of 30 days for the Venice (IT), Hornbæk (DK), Marseille (FR) and Newlyn (UK) sites.*

*Section 2.2.3, line 177. Please use consistent tenses throughout the paper when referencing other works. For instance, here you say "Bernardara et al. (2011) adopted", while previously you use the present tense (e.g. page 6, line 117 or later in the paper at page 8, line 212).*
Thank you, we have made verb tenses uniform throughout the manuscript.

*Page 9, line 239. I do not understand why the return period is expressed for annual maxima (AM). Apologies but I am not familiar with the MEVD, however it is clear that it allows to select multiple events per year. Then, why Eq. (3) is defined with respect to AM data?*
Exactly, the MEVD uses more observations to estimate the parameters of the distribution of "ordinary values". However, the MEVD cumulative distribution (Eq. (5) in the revised manuscript) is still the distribution of the annual maxima, estimated using a much greater sample than just yearly maxima. To clarify this point, the revised manuscript (lines 185-186) now reports: "Hence, the MEVD cumulative distribution of block maxima (estimated using a much greater sample than just yearly maxima used in the BM approach) is then defined as the compound probability: […]".

*It is not clear the purpose of Section 3.1, given that no non-stationary distributions are subsequently employed. However, if you want to keep it, I suggest adding the confidence intervals of the slopes fitted to the data (and perhaps comment them with respect to the p-values of the Mann-Kendall test).*
Section 3.1 is a preliminary analysis of the extreme sea levels to understand whether long-term trends, unrelated to sea-level rise and due to other factors (e.g., storm intensity and frequency, morphological variations, etc.), are present in level fluctuations from the mean.
To avoid any confusion about the purpose of this section, in the revised manuscript we have (1) changed the title of this section to "Mann-Kendall trend test", and (2) modified Figure 2 which now displays only the deviation of yearly maxima from mean sea level and 19-yr running mean. The revised section discusses here that:

"We start by computing mean sea level on yearly basis and by subtracting it from observed total water level. The first question that we want to explore is the presence of log-term trends, unrelated to sea-level rise and associated to other factors (e.g., human-induced factors, morphological variations, etc.), in the "cleaned up" signal, i.e. the observed measurements without mean sea level. To answer this question, in this work we focus on the deviation of yearly maxima from yearly mean sea level and test for the presence of trend by the two-tail Mann-Kendall test (Mann, 1945). Figure 2 summarizes results for each location explored. From a first visual inspection of Figure 2, the Venice (1872-2019) and Hornbæk (1891-2012) time series appear to show an increasing trend in the deviations of yearly maxima from yearly mean sea level (blue line) of different magnitudes. On the contrary, Marseille sea level observations (1985-2018) seem to be characterized by a decreasing trend. Finally, the Newlyn historical record (1915-2016) displays a fairly constant signal with no noticeable variations. The application of the Mann-Kendall test reveals a partly different story. The test rejects the hypothesis of the absence of trend, at the 95% confidence level, only for the Venice site (p-value$^{Venice}$ = 0.014). This result suggests that the increase of the yearly maximum deviations from yearly mean sea level may be a direct result of the local morphological variations of lagoon channels where the tidal wave propagates (whereby dissipation of the wave is reduced), and/or of land subsidence. On the contrary, at the remaining locations, the null hypothesis of no trend cannot be rejected (p-value$^{Hornbæk}$ = 0.352, p-value$^{Marseille}$ = 0.110, and p-value$^{Newlyn}$ = 0.997). The results obtained from these analyses support the hypothesis that mean sea-level rise is the dominant factor in determining the future frequency of coastal flooding (see §2.2.6). For the tests performed here to compare different extreme-value statistical models the possible presence of trends (e.g. in Venice) is irrelevant, since such tests are performed by first reshuffling observed values, thereby eliminating any existing trend, albeit small.".

*Section 3.2, lines 277-279. What does it mean that thresholds are selected "based on local tidal ranges"? This is a pivotal step of the study, please extend the explanation (you could also add it to the Methods section).*
Tidal range is defined as the sea level elevation between high and low water level over a tidal cycle. Tidal range may vary significantly across different sites. The local value of tidal range crucially determines the value of water level peaks, thus affecting the choice of the threshold.
In response to a comment by Reviewer#1 we have moved lines 276-277 (referred to the original version of the manuscript) to line 216 (referred to the revised version). To clarify, as requested by this Reviewer, the selection of the threshold, lines 213-218 now discuss that: "This threshold is chosen to be as small as possible (differently from the POT approach), to retain as much of the observational information as possible, and will be dependent on the magnitude of the local tidal range (sea level difference between high and low water level over a tidal cycle), and of storm contributions. Additionally, the threshold is set to be large enough to filter out water level peaks that are likely fully determined by tidal fluctuations, in the absence of any storm contribution. Given the above constraints, we also choose the threshold value that minimizes the estimation error under the MEVD framework".

*Figures S2-S6. Please use a 1:1 axis ratio. This would help to assess the quality of the fit.*
Thank you, agreed.

*Figure 5. Levels in the return period plots refer to z or h? I find the terminology rather confusing throughout the whole manuscript, e.g. sometimes you talk about storm surge, some other time about extreme sea levels. Please be consistent.*
We apologize for the lack of clarity. It is *z*. To avoid confusion, in Figure 7 (previously Figure 5) we have replaced "water level" with "total water level" (i.e. the variable *z*). In the revised manuscript, we have revised all the notation and terms used.

*Conclusions, line 349. Please specify that MEVD outperforms the other distributions for long enough calibration periods.*

Thank you for the suggestion. We have revised the sentence accordingly.

---

## Author Response (AR2)

**Authors' replies to Reviewers comments for NHESS-2021-236**

Once again we would like to thank Dr. Philip Ward and the two anonymous Reviewers for their helpful comments and suggestions, which further improved the manuscript.
Below we provide our discussion of Reviewer's comments (in blue italic font) and describe the changes addressing them (in black font).

**Reviewer #1:**

*"Extreme Coastal water level" is in the title but it is never mentioned in the abstract. In the abstract, the Authors refer to "extreme sea level" and "total water level". Moreover, projections are done on total water level (e.g., Section 2.2.6 and Fig. 7 – revised manuscript).*

Thank you for pointing out this inconsistency in the terminology used in the abstract. We have modified this section according to this comment, which it now reads: "Accurate estimates of the probability of extreme sea levels are pivotal for assessing risk and for designing coastal defense structures. This probability is typically estimated by modelling observed sea-level records using one of a few statistical approaches. In this study we comparatively apply the Generalized Extreme Value (GEV) distribution, based on Block Maxima (BM) and Peaks-Over-Threshold (POT) formulations, and the recent Metastatistical Extreme Value Distribution (MEVD) to four long time series of sea-level observations distributed along European coastlines. A cross-validation approach, dividing available data into separate calibration and test sub-samples, is used to compare their performances in high-quantile estimation. To address the limitations posed by the length of the observational time series, we quantify the estimation uncertainty associated with different calibration sample sizes, from 5 to 30 years. We study extreme values of the coastal water level – the sum of the water level setup induced by meteorological forcing and of the astronomical tide – and we find that the MEVD framework provides robust quantile estimates, especially when longer sample sizes of 10-30 years are considered. However, differences in performance among the approaches explored are subtle, and a definitive conclusion on an optimal solution independent of the return period of interest remains elusive. Finally, we investigate the influence of end-of-century projected mean sea levels, on the probability of occurrence of extreme total water levels (the sum of the instantaneous water level and the increasing mean sea level) frequencies. The analyses show that increases in the value of total water levels corresponding to a fixed return period are highly heterogeneous across the locations explored".

We believe that the use of "total water level" in the title, before an adequate definition of the term (given in Section 2.2.1 and, very succinctly, in the abstract) would be confusing to the reader.

*In the abstract, the Authors say that "the MEVD estimates outperform the traditional methods" (Line 10) while from the conclusion and the responses to reviewers it seems not possible to draw such a strong statement.*

In the revised manuscript, we have attenuated this conclusion. In particular, we have replaced this statement with: "We study extreme values of the coastal water level – the sum of the water level setup induced by meteorological forcing and of the astronomical tide – and we find that the MEVD framework provides robust quantile estimates, especially when longer sample sizes of 10-30 years are considered. However, differences in performance among the approaches explored are subtle, and a definitive conclusion on an optimal solution independent of the return period of interest remains elusive".

*Section 3.3. and discussion about the effect of changes in total water level as a function of mean sea level. I do appreciate the idea of trying to quantify the changes in Tr. However, changes in exceedance probability (so return period) depend mostly on the shape of the distribution (the tail) especially for high quantile while here the focus is on the frequency of the observations.*

We interpret this comment as asking to clarify the rationale for the discussions in Section 3.3. This section seeks to examine how events with different $Tr$ may change under sea-level rise at different sites. We believe that Eq. 9 serves this purpose. We have now further developed the discussion by highlighting how changes are site-dependent (lines 465-466): "Finally, Eq. 9 shows that percentage changes in $Tr$ are highly site-dependent through the shape of $f(z-msl)$."

*It is important that the Authors specify the value of msl used in Figure 7. The caption says "The green curve represents the estimates obtained with the observed record; the blue and red curves represent the estimates obtained with the projected sea-level rise (SLR) until 2100 with RCP4.5 (blue) and RCP8.5". Does it mean that msl is equal to mean sea level in the year 2100? The return level will differ significantly depending on msl assumed.*

We agree with this suggestion. As reported from lines 314 to 316, "2) we estimate the future probability of extreme total water levels by translating extreme level quantile estimates upward according to location-specific projections of mean sea level in the year 2100 […]"). Therefore, the *msl* is computed with respect to the year 2100. In the revised caption of Figure 7 we have changed "until 2100" with "in the year 2100".

**Reviewer #2:**

*As general comment: you state that the Poisson hypothesis is a limit of the POT-GPD approach, though it is known that the frequency of occurrence of the events does not affect significantly peak estimates (see e.g. Onoz & Bayazit, 2001). As such, the fact that the MEVD approach does not rely on such hypothesis really yields an advantage over the POT-GPD? please comment in the paper.*

We agree with this Reviewer that the literature shows that assumptions for the occurrence process that are different from the Poisson distribution lead to small differences in estimated quantiles. The relaxation of the Poisson assumption is not the main advantage afforded by the MEVD, and we have de-emphasized this point in the discussion at line 73 by removing "(while, e.g. POT-GPD assumes a Poisson occurrence process)". Önöz and Bayazit, (2001) were already cited in the manuscript.

*Page 8, line 202: please be more specific on the test used to select GPD also in view of the fact that you discuss the results at page 13, line 349.*

Lines 211-213 now report: "Based on the comparative evaluation of the performance of these distributions, e.g. using diagnostic quantile-quantile scatter plots, the Generalized Pareto distribution emerged as the best model for the "ordinary" coastal water level values".

*As you title Section 3.1 "Mann-Kendall trend analysis", you should at least mention the test in the Methods (perhaps in Section 2.2.1?).*

In the Methods section, we have added a brief introduction to the following Section 3.1 (lines 139-142): "[…] are identified and their values constitute the basis for subsequent analyses of (i) log-term trends study of maximum yearly departures from the average mean sea level (two-tail Mann-Kendall test, Mann (1945)), and (ii) statistical inference of past coastal flooding events and their potential future changes".

*Please review the grammar and some terminology. A few examples: […].*

Thanks, we have revised terminology and grammar throughout the manuscript.